# Establishment and characterization of new tumor xenografts and cancer cell lines from EBV-positive nasopharyngeal carcinoma

Weitao Lin [1], Yim Ling Yip [1], Lin Jia [1], Wen Deng[2], Hong Zheng [3,4], Wei Dai[3], Josephine Mun Yee Ko [3], Kwok Wai Lo [5], Grace Tin Yun Chung[5], Kevin Y. Yip [6], Sau-Dan Lee[6], Johnny Sheung-Him Kwan[5], Jun Zhang[1], Tengfei Liu[1], Jimmy Yu-Wai Chan[7], Dora Lai-Wan Kwong[3], Victor Ho-Fun Lee[3], John Malcolm Nicholls[8], Pierre Busson [9], Xuefeng Liu [10,11,12], Alan Kwok Shing Chiang[13], Kwai Fung Hui[13], Hin Kwok[14], Siu Tim Cheung [15], Yuk Chun Cheung[1], Chi Keung Chan[1], Bin Li[1,16], Annie Lai-Man Cheung[1], Pok Man Hau[5], Yuan Zhou[5], Chi Man Tsang[1,5], Jaap Middeldorp [17], Honglin Chen [18], Maria Li Lung[3] & Sai Wah Tsao[1]

The lack of representative nasopharyngeal carcinoma (NPC) models has seriously hampered research on EBV carcinogenesis and preclinical studies in NPC. Here we report the successful growth of five NPC patient-derived xenografts (PDXs) from fifty-eight attempts of transplantation of NPC specimens into NOD/SCID mice. The take rates for primary and recurrent NPC are 4.9% and 17.6%, respectively. Successful establishment of a new EBV-positive NPC cell line, NPC43, is achieved directly from patient NPC tissues by including Rho-associated coiled-coil containing kinases inhibitor (Y-27632) in culture medium. Spontaneous lytic reactivation of EBV can be observed in NPC43 upon withdrawal of Y-27632. Whole-exome sequencing (WES) reveals a close similarity in mutational profiles of these NPC PDXs with their corresponding patient NPC. Whole-genome sequencing (WGS) further delineates the genomic landscape and sequences of EBV genomes in these newly established NPC models, which supports their potential use in future studies of NPC.

[1] School of Biomedical Sciences, Li Ka Shing Faculty of Medicine, The University of Hong Kong, Hong Kong, China. [2] School of Nursing, Li Ka Shing Faculty of Medicine, The University of Hong Kong, Hong Kong, China. [3] Department of Clinical Oncology, Li Ka Shing Faculty of Medicine, The University of Hong Kong, Hong Kong, China. [4] Center for Biomedical Informatics Research, Stanford University, Stanford 94305 CA, USA. [5] Department of Anatomical and Cellular Pathology and State Key Laboratory of Translational Oncology, The Chinese University of Hong Kong, Hong Kong, China. [6] Department of Computer Science and Engineering, The Chinese University of Hong Kong, Hong Kong, China. [7] Department of Surgery, Li Ka Shing Faculty of Medicine, The University of Hong Kong, Hong Kong, China. [8] Department of Pathology, Li Ka Shing Faculty of Medicine, The University of Hong Kong, Hong Kong, China. [9] Gustave Roussy, Paris-Saclay University, CNRS, UMR8126, Villejuif F-94805, France. [10] Center for Cell Reprogramming, Department of Pathology, Georgetown University Medical Center, Washington 20057 DC, USA. [11] Department of Endocrinology and Metabolism, The Second Xiangya Hospital, Central South University, Changsha 410011 Hunan, China. [12] Affiliated Cancer Hospital & Institute, Guangzhou Medical University, Guangzhou 510095 Guangdong, China. [13] Department of Paediatrics and Adolescent Medicine, Li Ka Shing Faculty of Medicine, The University of Hong Kong, Hong Kong, China. [14] Center for Genomic Sciences, The University of Hong Kong, Hong Kong, China. [15] Department of Surgery and Li Ka Shing Institute of Health Sciences, Faculty of Medicine, The Chinese University of Hong Kong, Hong Kong, China. [16] College of Life Science and Technology, Jinan University, Guangzhou 510632 Guangdong, China. [17] VU University Medical Center, Department of Pathology, Cancer Center Amsterdam, de Boelelaan 1117, 1081 HV Amsterdam, The Netherlands. [18] Department of Microbiology, Li Ka Shing Faculty of Medicine, The University of Hong Kong, Hong Kong, China. These authors contributed equally: Weitao Lin, Yim Ling Yip, Lin Jia. Correspondence and requests for materials should be addressed to M.L.L. (email: mlilung@hku.hk) or to S.W.T. (email: gswtsao@hku.hk)

Nasopharyngeal carcinoma (NPC) is rare worldwide but common in southern China, including Hong Kong. The endemic NPC among southern Chinese is typically non-keratinizing carcinoma which is almost 100% associated with Epstein–Barr virus (EBV) infection[1].

Patient-derived xenografts (PDXs), given their close resemblance with patient tumors, serve as important models in pre-clinical evaluation for novel therapeutic drugs. For unclear reasons, it has been difficult to establish NPC PDXs in vivo. Currently, there are four NPC PDXs available for research, including X2117, C15, C17 and C18. However, all of them have been passaged in nude mice for over 25 years and may deviate from their original NPC tumors in patients[2,3]. In vitro, C666-1 is the only EBV-positive (EBV+ve) NPC cell line which has been used extensively in investigations. C666-1 was established from an NPC xenograft (X666) which had been propagated for a long period of time[4]. Most if not all the other previously reported NPC cell lines have lost their EBV episomes and became EBV-negative (EBV−ve) upon in vitro propagation[5,6]. Furthermore, many of these reported NPC cell lines have been shown with genetic contamination of HeLa cells[7,8]. Hence, their applications in NPC studies are limited. The scarcity of in vivo and in vitro NPC models represents major challenges for NPC and EBV research.

In this study, we report the successful establishment and comprehensive characterization of four new NPC PDXs (all EBV+ve) and three NPC cell lines (one EBV+ve; two EBV−ve). These newly established EBV+ NPC PDXs and cell lines significantly recapitulate the mutation profiles of their original NPC tumors, and harbor common genetic alterations reported in NPC, which supports their potential applications in the investigations of NPC pathogenesis. The newly established NPC PDXs can be propagated subcutaneously in NOD/SCID (non-obese diabetic/severe combined immunodeficiency) mice. Lytic EBV reactivation may be an intrinsic barrier to the successful establishment of EBV+ve NPC PDXs and cell lines. Inclusion of Y-27632, an inhibitor of Rho-associated coiled-coil containing kinases (ROCK), facilitated the establishment of a new EBV+ve NPC cell line, NPC43. NPC43 cells exhibited tumorigenicity in immunodeficient mice, and could be induced to undergo EBV lytic reactivation with production of infectious virions.

The establishment and characterization of new NPC PDXs and cell lines will provide valuable experimental tools for NPC and EBV research. Our experience in the establishment of these PDXs and cell lines will also facilitate future attempts to generate relevant and representative NPC models for investigations.

## Results

**Establishment of PDXs in immunodeficient mice.** In this study, attempts to establish NPC PDXs were initiated using 58 NPC patient samples, including 41 primary biopsies and 17 naso-pharyngectomized recurrent tumors. Subrenal implantation of NPC specimens was performed in NOD/SCID mice, and examined for growth after 4 to 6 months. Five NPC xenografts exhibited signs of growth, including Xeno23, 32, 43, 47 and 76 (Fig. 1a). Four of these xenografts (Xeno23, 32, 47 and 76) exhibited subcutaneous growth in NOD/SCID mice, and could be transplanted and propagated accordingly (Fig. 1b). Multiple transfers of NPC xenografts to new mice were usually required before robust growth of the transplanted xenografts could be observed. In the case of Xeno23, stable growth of transplanted PDX was only observed after the seventh transfer in mice (Supplementary Fig. 1a). Unfortunately, very limited growth of Xeno43 was observed after transfer to new mice, which was eventually lost after the fifth transfer (Supplementary Fig. 1b).

The detailed clinical information of NPC samples with successful establishment of PDXs is shown in Table 1. Xeno32 and 76 were derived from NPC primary biopsies, while Xeno23, 43 and 47 were from surgically resected recurrent NPC. Notably, all recurrent NPC cases, including cases 23, 43 and 47, are free from regional lymph node and distant metastases, indicating they are developed from a primary NPC. A higher take rate was observed from surgically resected recurrent NPC tissues (3/17 cases; 17.6%) compared to primary biopsies (2/41 cases; 4.9%). Despite the failure in maintaining Xeno43, we established a new EBV+ve NPC cell line, NPC43, in vitro directly from patient NPC tissue. The details of establishment and characterization of NPC43 will be described in the later section of this report.

The origins of all the newly established NPC PDXs from patients were confirmed by short tandem repeat (STR) profile analysis. Besides, their STR profiles were distinct from the currently available NPC PDXs which have been passaged for a long time (Supplementary Table 1).

The epithelial origin and presence of EBV infection in these newly established PDXs were confirmed by immunohistochemical staining using pan-keratin antibodies (AE1 and AE3) and EBER (EBV-encoded RNA) in situ hybridization (ISH), respectively (Fig. 1c). All the established PDXs, including Xeno43, showed positive keratin and EBER expression. The epithelial nature of the four established NPC PDXs was further confirmed by the presence of desmosomes by transmission electron microscopy examination (Fig. 1d).

**Lytic EBV reactivation in PDXs.** For unclear reasons, the overall success rate of establishment of transplantable and maintainable NPC PDXs in this study was low (4/58 cases; 6.9%), compared to that from other head and neck cancers[9]. We examined some clinical properties of NPC which might affect the establishment, including clinical status and outcome of patients (Table 1 and Supplementary Table 2). However, no significant correlation between success rate and these clinical factors was observed. We further examined the tumor contents in tissues and the plasma EBV copy number in patients in the available cases and still found no clear correlation with the success rate of PDX establishment (Supplementary Table 3).

We next performed RNA-ISH by RNAscope® analysis platform to examine the messenger RNA (mRNA) expressions of key EBV lytic genes in the newly established PDXs (Fig. 2a). Expression of BZLF1, BRLF1, BMRF1 and BLLF1 was indicated by clusters of hybridization signals in the newly established PDXs. However, hybridization signals of lytic transcripts were not observed in long-term passaged C15, or NPC specimens (NPC1 and NPC2) from patients. Higher expression of lytic EBV genes in the newly established PDXs (Xeno23, 32, 47 and 76) was also revealed by real-time PCR when compared with that in long-term passaged PDXs (C15, C17, X666 and X2117) (Fig. 2b). Furthermore, lytic EBV genes also exhibited higher expression in the earlier passages of PDXs (Xeno32, 47 and 76) as compared to that in their respective later passages (Supplementary Fig. 2). Apparently, there is a selection for latently EBV-infected NPC populations during their propagation in mice. These observations support our hypothesis that lytic reactivation of EBV in NPC tumors transplanted to immunodeficient mice may lead to low success rate of PDX establishment.

**Establishment of a new EBV+ve NPC cell line.** We also attempted to establish NPC cell lines from patient NPC specimens. No success in cell line establishment was achieved from the initial 13 attempts using RPMI-1640 medium supplemented with 10% fetal bovine serum (FBS). Epithelial outgrowths were

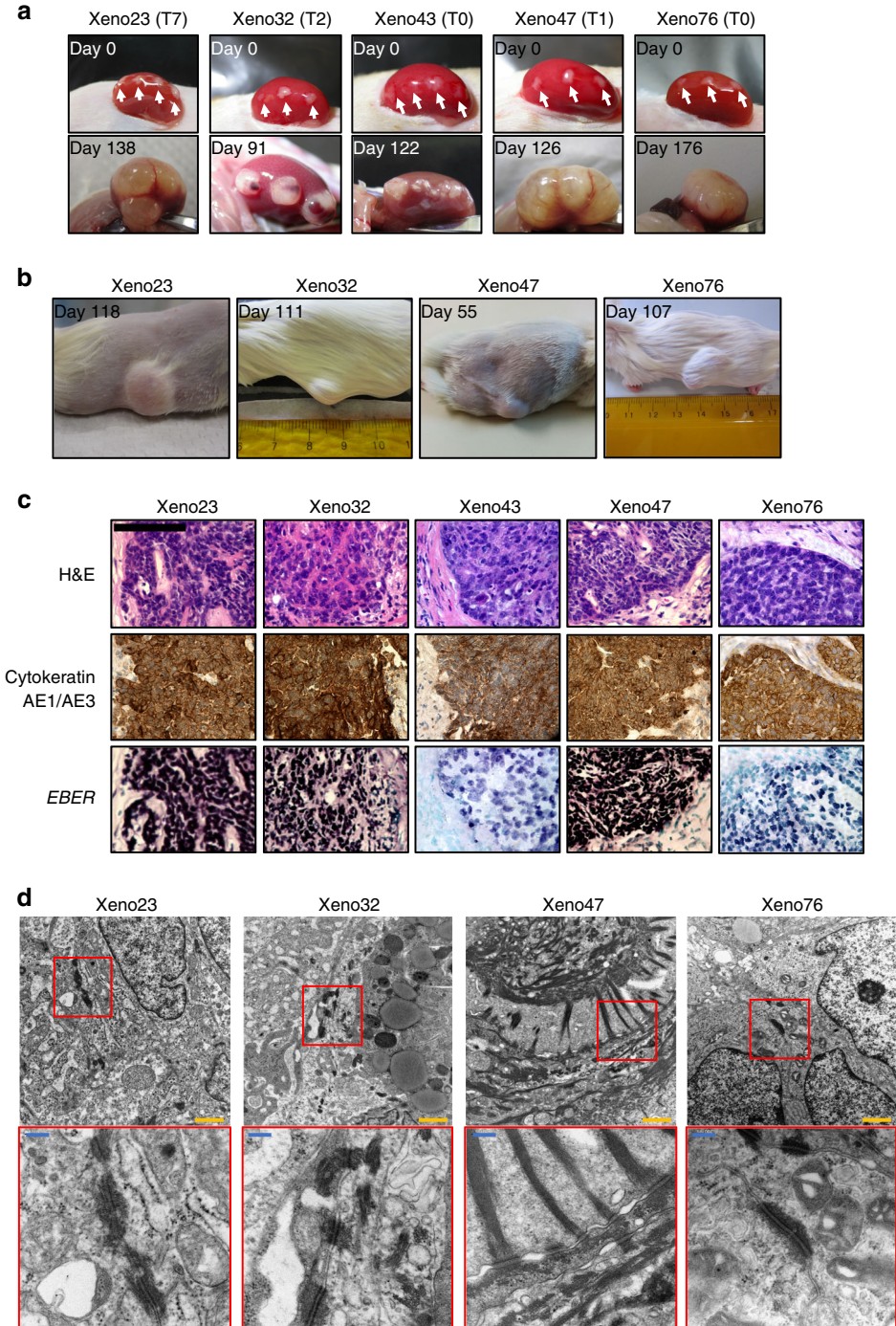

**Fig. 1** Establishment of NPC PDXs in immunodeficient mice. **a** Upper panels: Establishment of NPC PDXs (Xeno23, 32, 43, 47 and 76) was achieved by implantation of patient NPC tissues (white arrows) under kidney capsules in NOD/SCID mice. Lower panels: Growth of PDXs observed in the same mice at 138, 91, 122, 126 and 176 days respectively after implantation. **b** The growth of NPC PDXs (Xeno23, 32, 47 and 76) was observed in NOD/SCID mice at 118, 111, 55 and 107 days after subcutaneous implantation. **c** H&E, immunohistochemical staining of cytokeratin AE1/AE3 and *EBER* ISH were performed in consecutive sections of all five NPC PDXs. The results of H&E staining confirmed that the PDXs were undifferentiated NPC. The epithelial nature of these PDXs and the presence of EBV were confirmed by the expression of keratin and *EBER*, respectively. Scale bar, 100 μm. **d** The presence of desmosomes and different abundance of tonofilaments in four PDXs (Xeno23, 32, 47 and 76) as revealed by transmission electron microscopy, confirming the epithelial origin of the tumor cells. Scale bars: 1 μm (yellow); 200 nm (blue)

observed in 4 cases, but none of them could be expanded as continuous culture. We postulated that lytic EBV reactivation might be a barrier for successful establishment of cell line. ROCK inhibitor Y-27632 has been reported with suppressive functions in the differentiation of squamous epithelial cells and promotes the establishment of continuous cell lines from multiple types of

human tumors[10]. Recent evidence also demonstrated its effects in suppressing tetradecanoyl phorbol acetate (TPA)-induced EBV lytic replication[11]. We then examined whether Y-27632 could facilitate NPC cell line establishment. We observed a comparable rate of epithelial outgrowth from NPC explants in Y-27632-containing culture medium (11 out of 33 cases); however, 3 of

**Table 1 Clinical data of donor patients**

| Case no. | Sex | Age | Histological diagnosis[a] | Clinical status | | | | Sample type[b] | Tumor recurrence | Tumor metastasis | History of chemotherapy[c] | History of RT treatment |
|---|---|---|---|---|---|---|---|---|---|---|---|---|
| | | | | T | N | M | Overall staging | | | | | |
| 23[d] | M | 65 | Non-keratinizing carcinoma | 2 | 0 | 0 | II | Nasopharyngectomized tissue | Yes | No | No | 2D-RT (66 Gy in 33 fractions) |
| 32 | F | 72 | Undifferentiated carcinoma | 3 | 2 | 0 | III | Primary biopsy | No | No | ChemoRT then adjuvant chemotherapy | IMRT (70 Gy in 33 fractions) |
| 38[d] | F | 63 | Moderately differentiated squamous cell carcinoma | 2 | 0 | 0 | II | Nasopharyngectomized tissue | Yes | No | No | 2D-RT (66 Gy in 33 fractions) |
| 43[d] | M | 64 | Poorly differentiated carcinoma | 3 | 0 | 0 | III | Nasopharyngectomized tissue | Yes | Yes | ChemoRT then adjuvant chemotherapy | 2D-RT (66 Gy in 33 fractions) |
| 47[d] | F | 52 | Undifferentiated carcinoma | 3 | 0 | 0 | III | Nasopharyngectomized tissue | Yes | No | ChemoRT alone | IMRT (70 Gy in 35 fractions) |
| 53[d] | M | 39 | Undifferentiated carcinoma | 4 | 1 | 0 | IVA | Nasopharyngectomized tissue | Yes | No | ChemoRT alone | IMRT (70 Gy in 35 fractions) |
| 76 | F | 57 | Undifferentiated carcinoma | 1 | 0 | 0 | I | Primary biopsy | Yes | No | No | IMRT (66 Gy in 33 fractions) |

[a]Histopathological properties of the NPC from which PDXs and cell lines were successfully established (from the original report from pathologists in Queen Mary Hospital)
[b]Patients with newly diagnosed NPC received either curative RT or ChemoRT with or without adjuvant chemotherapy at the time of diagnosis. When NPC recurred, patients received nasopharyngectomy. Therefore, primary biopsy was obtained before any treatment, while nasopharyngectomized tissue was recurrent tumor and collected after treatment
[c]Patients who had ChemoRT received cisplatin 100 mg m$^{-2}$ every 3 weeks for 3 cycles. For those who received adjuvant chemotherapy after ChemoRT, they received cisplatin 80 mg m$^{-2}$ and 5-FU 1000 mg m$^{-2}$ every 4 weeks for 3 more cycles
[d]Patients who did not survive by 30 June 2018

these cases (NPC38, 43 and 53) could be expanded and propagated. Real-time PCR confirmed the presence of EBV in NPC43 but not in NPC38 (0 copy of EBV in PD 71) and 53 (0 copy of EBV in PD 33).

The detailed clinical information of patients 38, 43 and 53 is included in Table 1. All three NPC cell lines were established from nasopharyngectomized recurrent NPC tissues. Patient specimen of NPC38 shows histological features of squamous cell carcinoma, and no EBV infection by *EBER* staining (Supplementary Fig. 3a). Hence, NPC38 may represent an independent primary squamous cell carcinoma induced by radiotherapy at the recurrent site of NPC. The patient specimens of NPC43 and NPC53 were *EBER*-positive (Fig. 3a and Supplementary Fig. 3b). Intriguingly, low average EBV copy number (0.001098 ± 0.000225 EBV copy per cell) was detected in the first passage of NPC53. Given the *EBER* positivity in patient NPC specimen as well as the presence of EBV in NPC53 cell line at early passage, NPC53 cell line probably represents an NPC cell line which lost its EBV episomes during establishment in culture. Notably, a long period of time (>500 days) was required for NPC53 to reach confluency before the first subculture (Supplementary Fig. 3c). During this long period of culturing time, EBV−ve NPC53 cells may outgrow EBV+ve NPC53 cells and become the dominant cell type in culture. STR profiles of NPC38, 43 and 53 compared with their corresponding patients' blood DNA confirmed their origins (Supplementary Table 1).

The procedures and experiences for establishment of the new EBV+ve NPC43 cell line are described in detail here. Epithelial outgrowths from multiple NPC43 explants were observed within 1 week in the primary culture (Fig. 3b). Only in the presence of Y-27632 (4 μM), the outgrowths continued to expand, and could be subcultured after 54 days (Fig. 3c). The splitting ratio of NPC43

was kept low at 1:2 for early passages. After 22 population doublings (PDs), a higher split ratio (1:4) was used. The NPC43 cells have been subcultured over 100 times, achieving total PDs of over 200 without any signs of senescence (Fig. 3c). The mean PD time of NPC43 was estimated to be 8, 4 and 2.5 days at PD 22, 90 and 200, respectively, by the growth curve. An independent proliferation assay based on thymidine incorporation also confirmed an increased proliferation rate of NPC43 at later passages (Supplementary Fig. 4).

NPC43 cells at different passages (PD 2, 17 and 108) were authenticated by comparing the STR profiles with the profile of patient blood DNA (Supplementary Table 1). The epithelial origin of NPC43 cells was confirmed by the expression of cytokeratin (Fig. 3d). EBV genomes in NPC43 were identified by fluorescent ISH (FISH) analysis using EBV-specific DNA probes (Fig. 3e). EBV latent and lytic gene expression in NPC43 at PD 132, including *EBNA1*, *EBER1/2*, *LMP1*, *BZLF1* and *BRLF1*, was examined by real-time PCR (Fig. 3f). Tumorigenicity of NPC43 was demonstrated in NOD/SCID mice 3 months after subcutaneous injection of $10^7$ cells (Fig. 3g). Histological examination of the tumor developed from NPC43 cells confirmed its undifferentiated features with the presence of EBV by hematoxylin and eosin (H&E) and *EBER* staining (Fig. 3g).

Spectral karyotyping of NPC43 revealed a near-diploid karyotype with extensive chromosomal abnormalities, including loss of chromosomes 3, 8, 13, 14, 16, 21, 22, Y and gain of chromosome 19 (Fig. 3h). Chromosome losses of 3p, 13q, 14q and 16q have been reported and shown with functional significance in NPC carcinogenesis in earlier studies[12–15].

Most if not all the reported NPC cell lines except C666-1[4] and C17[16] eventually lost their EBV episomes during in vitro culture[5,6]. We next examined whether NPC43 could retain EBV

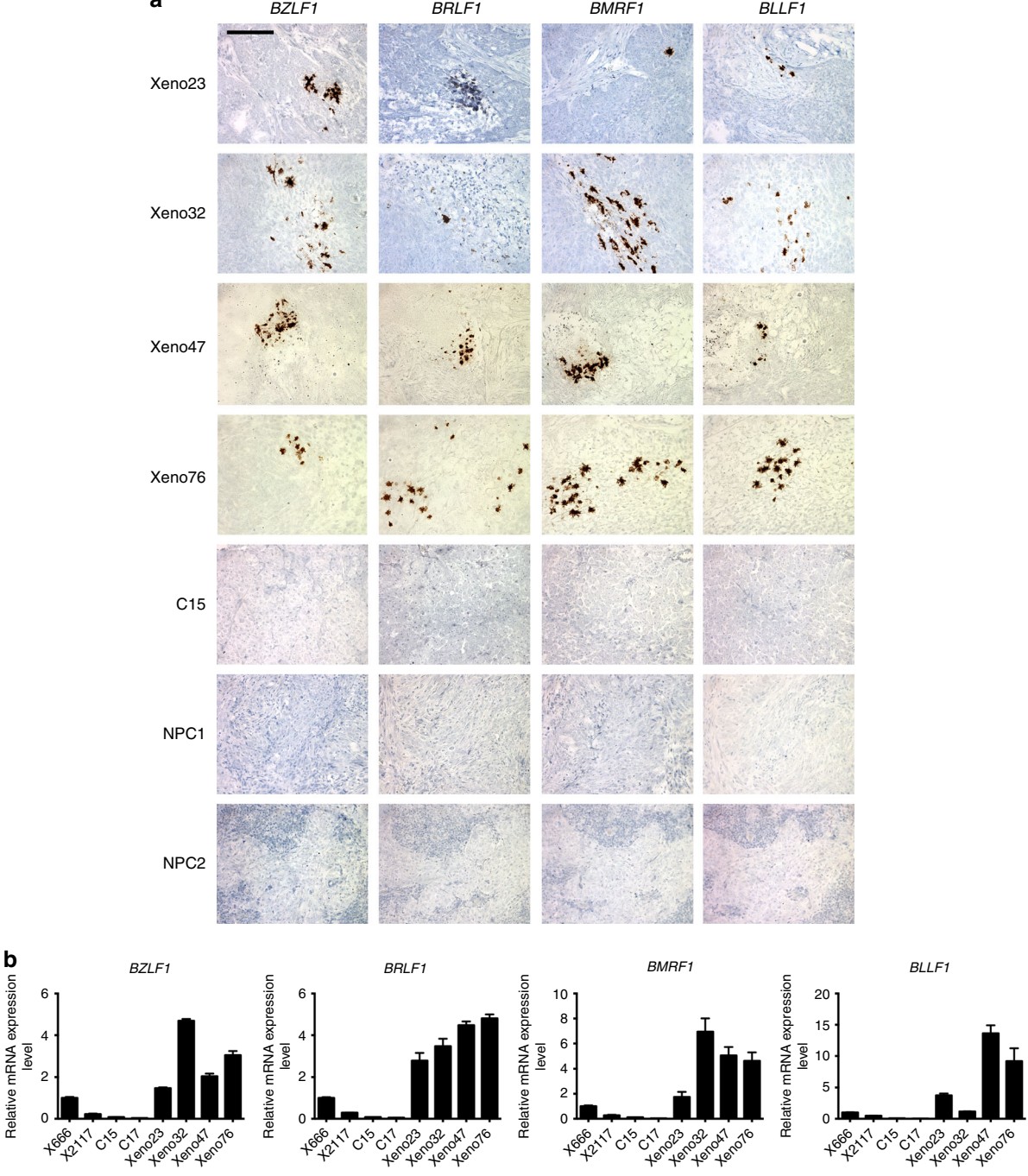

**Fig. 2** Expression of lytic EBV genes in NPC PDXs. **a** The expression of *BZLF1*, *BRLF1*, *BMRF1* and *BLLF1* transcripts was determined by RNAscope® in situ hybridization. Positive signals are shown as brown dots. Clusters of hybridization signals over NPC cells were interpreted as cells undergoing active lytic EBV infection. Abundant expression of lytic EBV genes was found in all newly established PDXs, but not in C15 or human NPC specimens (NPC1 and NPC2). Scale bar, 100 μm. **b** Expression of EBV lytic genes in four long-term passaged and four newly established NPC PDXs (Xeno23, 32, 47 and 76) quantified by real-time PCR. The newly established PDXs exhibited higher levels of lytic gene expression. Data are shown as mean ± SD from three independent experiments

episomes during its propagation. Average EBV copy number of NPC43 was examined at different passages by real-time PCR. A gradual decrease of average EBV copy number of NPC43 was observed during propagation at early passages (from 100 copies at PD 5 to 34 copies at PD 22). It became relatively stable after PD 26, with 15–20 copies per cell, and eventually stabilized as ~10 copies per cell after PD 100 (Fig. 4a). FISH analysis also confirmed the dynamic profile of EBV copy numbers in NPC43 cells at early and late subcultures (Supplementary Fig. 5).

**ROCK inhibitor suppressed lytic reactivation of EBV in NPC43.** To determine the involvement of ROCK inhibitor (Y-27632) in the suppression of lytic EBV reactivation in NPC43, EBV copy number in NPC43 at early passage (PD 10) culturing in different concentrations of Y-27632 was examined (Fig. 4b). Upon removal of Y-27632, average EBV copy number increased to 417 ± 9 per cell, which was around 4 times in NPC43 cells cultured with 4 μM Y-27632. Decreased average EBV copy numbers were also observed in NPC43 treated with higher

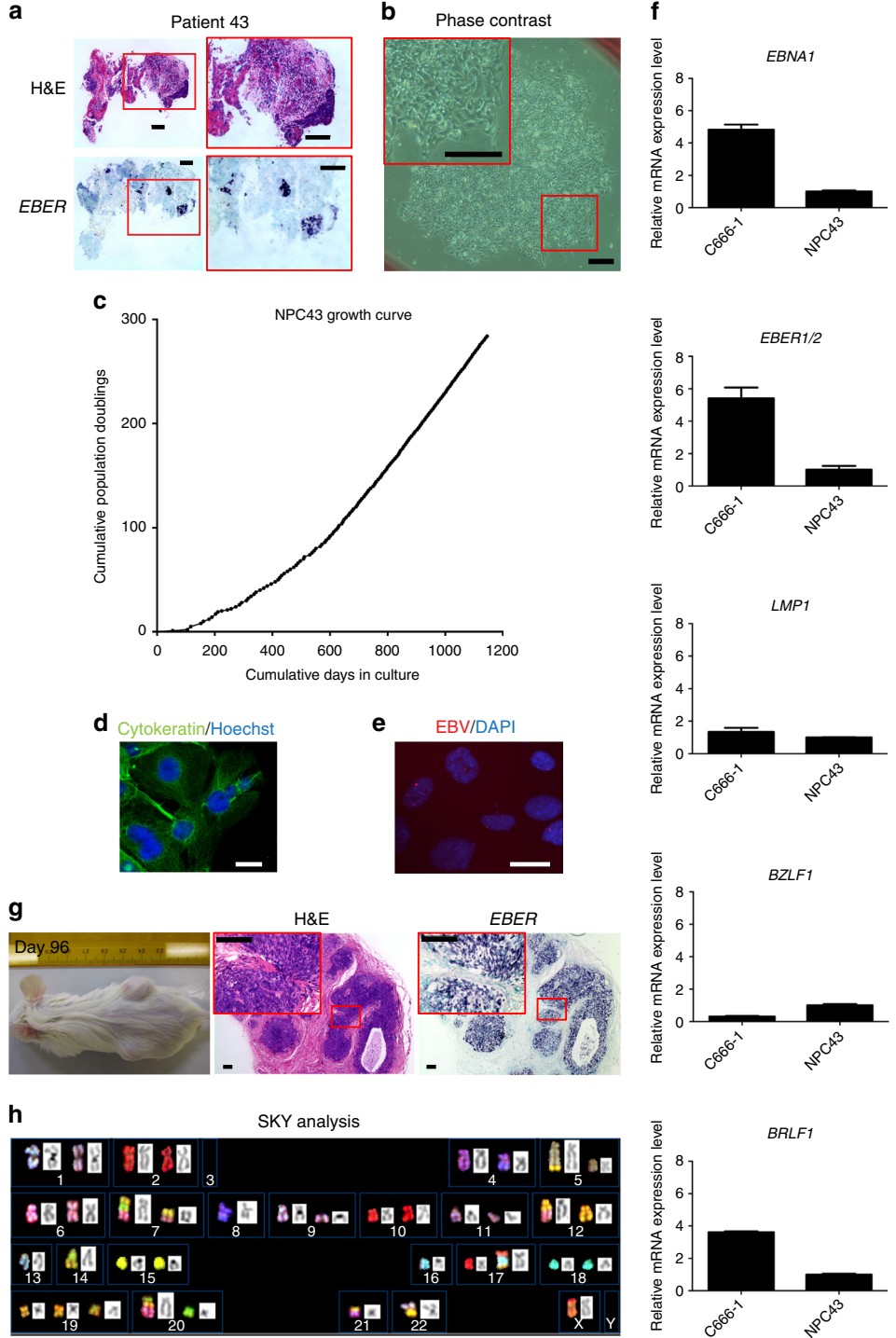

concentrations (10 and 20 μM) of Y-27632. We hypothesized that the higher EBV copy number detected in Y-27632-free medium was contributed by lytic EBV reactivation in a subpopulation of NPC43 cells. We next examined the expression of lytic EBV proteins (Rta, Zta, BALF5, EA-D and Gp350/220) by western blotting in NPC43 cells cultured in the presence or absence of Y-27632 (Fig. 4c). Expression levels of lytic EBV proteins in NPC43 cells diminished with increasing concentrations of Y-27632 and were completely suppressed by Y-27632 at 20 μM. The percentages of NPC43 cells expressing EBV lytic proteins with the treatment of different concentrations of Y-27632 were also examined by immunofluorescence (IF) staining (Fig. 4d).

Expression of Zta protein was detected in 9.56% of NPC43 cells upon withdrawal of Y-27632. The percentage of Zta-expressing NPC43 cells decreased in a dose-dependent manner with increasing concentrations of Y-27632 (2.1%, 0.6% and 0% at 4, 10 and 20 μM of Y-27632, respectively). Similar expression trends of two other lytic EBV proteins (EA-D and BALF2) were also observed. Hence, our results confirmed that Y-27632 effectively suppressed lytic EBV reactivation in culturing NPC43 cells at early passages. NPC43 at late passage (PD 280) showed less sensitivity to the EBV lytic induction by removal of Y-27632 (Supplementary Fig. 6), indicating that NPC43 at its later passage became less dependent on Y-27632. These results also support

**Fig. 3** Establishment and characterization of NPC43 cell line. **a** Histological examination of NPC tissue from patient 43. Although a low tumor content identified, H&E and *EBER* ISH staining showed the presence of undifferentiated and EBV+ve cancer cells in the tissue. Scale bars, 100 μm. **b** The phase-contrast micrograph shows the outgrowth of epithelial cells from the explanted NPC tissues from patient 43 at Day 7. Scale bars, 200 μm. **c** Growth curve of NPC43 cell line. The growth of NPC43 cells from the NPC explants was slow in the beginning and took about 50 days to become confluent for the first two passages (1:2 splitting ratio). After PD 22, the NPC43 cells could be passaged at a 1:4 splitting ratio. The mean doubling time of NPC43 cells was about 8, 4 and 2.5 days at PD 22, 90 and 200, respectively. **d** NPC43 cells were stained for the expression of cytokeratin AE1/AE3 (green). The nuclei were revealed by Hoechst stain (blue). The epithelial nature of NPC43 was confirmed by the positive staining of pan-cytokeratin in cells using specific antibodies. Scale bar, 20 μm. **e** Images of EBV FISH revealing the presence of EBV in NPC43 as punctate red dots. Scale bar, 20 μm. **f** Expression of EBV genes was detected in NPC43 by real-time PCR. C666-1 was used as positive control. Data are shown as mean ± SD from three independent experiments. **g** Left panel: Representative image of NPC43 tumors subcutaneously grown in NOD/SCID mouse. Tumors were harvested for histological examination 96 days after injection of NPC43 cells. Middle panel: H&E staining shows the presence of undifferentiated tumor cells. Right panel: *EBER* ISH confirmed that these tumor cells were EBV+ve. Scale bars, 100 μm. **h** A representative spectral karyotype of a metaphase spread prepared from NPC43 (PD 24). The karyotype of NPC43 exhibits complex chromosomal abnormalities. Karyotype description: 38,X,-Y,i(1)(q10),-3,-3, der(4)t(4;8)(q21;q?), der(5)t(5;11) (q35;?)t(11;12)(?;?),i(5)(p10),i(6)(q10),der(7)t(7;20)(q11;?)t(3;20)(q13;?),der(7)t(3;7)(?;q11),-8,del(9)(p11),del(11)(q11), del(11)(p11),der(12)t(3;12)(?;q13), -13,-14,der(14)t(7;14)(q?;q21),-16,der(17)t(16;17)(q10;q11)t(12;16)(?;?),+19,der(20)t(12;20)(q?;q11)t(3;12)(?;q13),-21,der(21)t(12;21)(p11,p11),-22, der(1;22) (?;p11) [cp12]

our hypothesis that lytic EBV reactivation in NPC cells interferes with the establishment of EBV+ve NPC cells in vitro.

**Lytic reactivation of NPC43 produces infectious EBV virions**. We next examined whether NPC43 cells would show responsiveness to EBV lytic induction by TPA treatment. The expression of lytic EBV proteins, including Rta, Zta and EA-D, could be detected in NPC43 (PD 102) after treatment with TPA for 48 h (Fig. 5a). To evaluate the proportion of cells that were responsive to TPA-mediated EBV lytic induction, IF staining using antibodies against Zta, EA-D and BALF2 was performed. The results indicated that a small percentage (1 to 2%) of NPC43 cells at PD 68 could be induced to undergo EBV lytic reactivation upon TPA treatment (Supplementary Fig. 7). We also examined whether infectious virions could be produced by NPC43 cells upon EBV lytic induction using the procedures illustrated in Fig. 5b. Briefly, the supernatant from NPC43 cells induced to EBV lytic replication was harvested for co-culture with EBV−ve Akata cells and human primary B cells. As shown in Fig. 5c, infection of NPC43-EBV in Akata cells could be verified by EBV DNA FISH. Besides, expressions of EBV latent and lytic genes were characterized in NPC43-EBV-infected Akata cells. The supernatant collected from lytic-induced HONE1-EBV cells was used as the positive control to infect EBV−ve Akata cells. Since the EBV virions produced by HONE1-EBV cells were green fluorescent protein (GFP)-tagged, the infected Akata cells became GFP-positive, which suggested the feasibility of this method. Comparable levels of lytic and latent EBV gene expression were detected in Akata cells infected by supernatants harvested from NPC43 and control HONE1-EBV cells induced to undergo lytic EBV infection (Fig. 5d). Furthermore, the supernatant harvested from NPC43 was also used to infect human primary B cells. Although at low rate, B transformation by NPC43-EBV infection could be detected, which further demonstrated the capacity of infectious virion production of NPC43 cells upon lytic induction (Fig. 5e). The EBV copy number in transformed B cells was determined by real-time PCR as $7.22 \times 10^4$ copies per ng DNA. By single-cell sorting, several EBV+ve Akata clones were generated. As illustrated in Supplementary Fig. 8, two representative EBV+ve Akata clones both showed decreased EBV copy number after single-cell sorting, suggesting NPC43-EBV infection in Akata cells might not confer growth advantage in vitro.

**Genetic landscapes of newly established NPC tumor lines**. Mutational profiles of PDXs (Xeno23, 32, 47) and cell line (NPC43) was compared with their corresponding patient tumor DNA by whole-exome sequencing (WES) analysis.

A total of 269 non-silent somatic mutations, including missense, stopgain, splicing, insertions and deletions (INDELs), in 261 genes were identified in four pairs of patient NPC and their derived PDXs/cell line (Fig. 6a). The overlap of mutations between NPC PDXs/cell line and their corresponding patient NPC tumors was 82% in Xeno23, 81% in Xeno32, 64% in Xeno47, and 94% in NPC43 cell line (Fig. 6b).

Four cancer-relevant genes, *NRAS*, *TP53*, *EP300* and *SMG1*, showed recurrent mutations in these PDXs/cell line by WES data analysis. Recurrent somatic hotspot mutations of *NRAS* were identified (Gln61Lys in Xeno23; Gln61Arg in Xeno32), leading to the activated forms of NRAS. These missense mutations have been reported in multiple types of human malignancies, including melanoma, colorectal, lung and thyroid tumors[17,18]. Sanger sequencing verified the *NRAS* mutations (Supplementary Fig. 9). WES also revealed somatic mutations of *TP53* in Xeno32 and NPC43 (Gly245Asp in Xeno32; Trp53* in NPC43), which were further verified by Sanger sequencing (Supplementary Fig. 10). Recurrent mutations of *TP53* in NPC have been previously reported[19,20]. *EP300* mutations were recurrently found in Xeno32 and NPC43. As one of the chromatin modifiers, inactivating mutations in *EP300* have been implicated in many human cancer types[21]. *SMG1* gene was mutated in both Xeno23 and 47. Sanger sequencing in Xeno23 verified the mutation (Supplementary Fig. 11). According to a recent report, SMG1 can suppress CDK2 and, thereby, regulate tumor growth through cell cycle regulatory pathways of p53 and cdc25A[22].

Whole-genome sequencing (WGS) analysis was performed in the established NPC PDXs and cell lines to further examine somatic mutations, including single-nucleotide variants (SNVs), INDELs, structural variants (SVs) and copy number variants (CNVs) (Supplementary Data 1–3). The Circos plots show the genomic features of each EBV+ve PDX and cell line (Fig. 7a; Supplementary Figs. 12–16).

As illustrated in Fig. 7b and Supplementary Data 3, copy number analysis revealed consistent chromosomal deletion in the regions of chromosomes 3p, 14q and 16q in all the EBV+ve NPC PDXs and NPC43 cell line. In addition, frequent loss of chromosomes 5q and 9p and gain of chromosome 1q were detected in 4 out of 5 cases. Multiple amplicons including 9p24, 11q13, 12p12-ter, 15q11-14 and 16p13 were also detected in these lines. Hence, the chromosomal aberration profiles of the newly established NPC PDXs and NPC43 cell line were consistent with those reported in patient NPC[12,20]. As illustrated in Fig. 8a, the common somatic alterations targeting the cell cycle regulation in

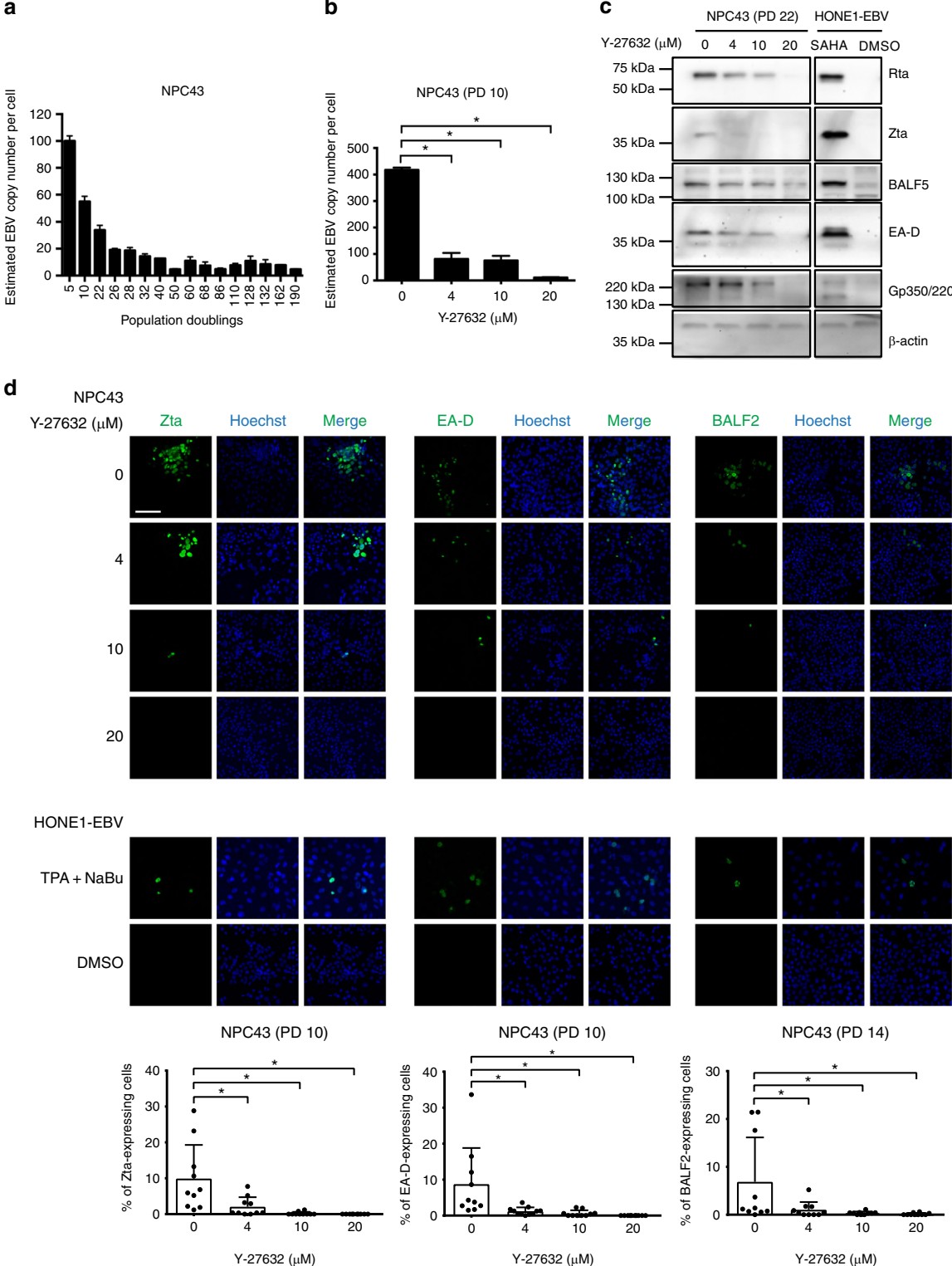

NPC are the homozygous deletion of 9p21.3 including the *CDKN2A/CDKN2B* loci in three PDXs (Xeno23, 47 and 76) and amplification of *CCND1* on 11q13 in NPC43 cell line. Over-expression of cyclin D1 has been shown in our previous study to support stable EBV infection in nasopharyngeal epithelial cells[23]. In addition, a unique 9p24 amplicon harboring the *JAK2*, *CD274* (*PDL1*) and *PDCD1LG2* (*PDL2*) genes was found in Xeno23, and homozygous deletions of *TGFBR2* on 3p24 were found in Xeno47 and NPC43. Their involvement in NPC pathogenesis remains to be further elucidated.

A total of 508 SVs were detected in the EBV+ve PDXs and cell line (Supplementary Data 2). The highest frequency of intra- and inter-chromosomal rearrangements was detected in NPC43 (Fig. 7a). Abundant SNVs and CNVs were detected in NPC43 cells suggesting genomic instability in this newly established NPC cell line. *CYLD* is a critical negative regulator of nuclear factor (NF)-κB pathways frequently mutated in NPC[19,20]. As shown in Fig. 8a, mutations of *CYLD* were commonly detected in EBV+ve NPC PDXs and cell line. Recurrent SVs of *CYLD*, including deletion, tandem duplication,

**Fig. 4** Spontaneous lytic EBV reactivation in NPC43 during cell line establishment. **a** Changes of average EBV copy number in NPC43 cells during establishment and propagation determined by real-time PCR. Estimated EBV copy number per cell at PD 5 was about 100. Gradual decrease of EBV copy number was observed in NPC43 during propagation from PD 5 to 26. EBV copy number became relatively stable from PD 26 to 190 (around 10–20 per cell). Data are shown as mean ± SD from three independent experiments. **b** NPC43 at PD 10 was treated with different concentrations of Y-27632 and the EBV copy number per cell was estimated by real-time PCR. In the absence of Y-27632, the EBV copy number per cell increased to 417 which may be due to induction of lytic reactivation of EBV in infected NPC43 cells. The copy number was around 100 at the concentrations of 4 and 10 µM. When the concentration of Y-27632 was increased to 20 µM, the copy number was around 10–20 per cell; *$p < 0.05$ in a two-tailed $t$-test. Data are shown as mean ± SD from three independent experiments. **c** Expression of EBV lytic proteins in NPC43 (PD 22) cultured under different concentrations of Y-27632. Expression levels of lytic proteins (Rta, Zta, BALF5, EA-D and Gp350/220) was negatively correlated with the concentration of Y-27632. Expression of β-actin was the loading control. HONE1-EBV treated with SAHA was included in this western blotting as a positive control for induction of lytic EBV reactivation. **d** IF staining results revealing the expression of lytic EBV proteins. The expression of Zta, EA-D and BALF2 was examined in NPC43 cells at early passages cultured in different concentrations of Y-27632 (top panel). HONE1-EBV treated with TPA and NaBu was included in the IF staining as a positive control of lytic EBV reactivation (middle panel). Quantification of percentage of positive-stained cells was performed by cell number counting. Data are shown as mean ± SD from 10 different microscopic views with over 2000 cells included; *$p < 0.05$ in a two-tailed $Z$-test. Scale bar, 100 µm

and inversion, were detected and verified in three of the newly established NPC PDXs/cell (Xeno23, 47 and NPC43) (Fig. 8a; Supplementary Fig. 17). Besides, a homozygous nonsense mutation (Ser371*) of *CYLD* was found and verified in Xeno76. In addition to *CYLD*, inactivation of other negative regulators of NF-κB pathways, including *TRAF3* and *BIRC2* (Fig. 8a; Supplementary Fig. 18), were observed in NPC43 and Xeno32 by homozygous frameshift mutation and translocation, respectively. These findings implicate that somatic alterations targeting NF-κB signaling pathway are common in these NPC PDXs and cell line, which confirms with the frequent somatic mutations of these negative regulators of NF-κB signaling pathways detected in clinical NPC tumors[20]. Intriguingly, as an EBV–ve cell line, NPC53 harbors somatic mutations in *CYLD* and *TRAF3* as EBV+ve PDXs and cell lines, which is totally distinct from the genetic mutation landscape of NPC38. As discussed earlier, NPC53 may represent an originally EBV-infected NPC cell line which subsequently lost its EBV episomes upon propagation in culture. The exclusive mutation profiles of EBV+ve and EBV–ve PDXs and cell lines may indicate the difference in their driving forces in carcinogenesis as well as susceptibility and maintenance of EBV infection.

Besides NF-κB pathway, we also identified missense mutation of *NRAS* in Xeno23, 32 and C666-1 as well as a homozygous missense mutation of *PTEN* in Xeno47 (Fig. 8a; Supplementary Fig. 19), which further suggests an aberrantly activated phosphoinositide-3-kinase (PI3K)/AKT signaling pathway in EBV+ve NPC PDXs and cell lines. These mutations have been reported in an earlier NPC genomic study[24].

The transcriptome profiles of the newly established NPC PDXs and cell lines were also examined to explore whether differential gene expressions could be detected in EBV–ve and EBV+ve cohorts. A detailed summary of sequencing data and mapping is included in Supplementary Table 4. By quantification and comparison of gene expression levels, 1974 differentially expressed genes were identified between EBV–ve and EBV+ve cohorts (Supplementary Fig. 20). By gene set enrichment analysis (GSEA), a significant enrichment in NF-κB and PI3K pathways was revealed with gene upregulation in EBV+ve cohorts, which is consistent with the genetic alterations in the regulators of NF-κB and PI3K pathways identified by WGS (Fig. 8b; Supplementary Fig. 21a). A panel of NF-κB targets exhibited increased expression pattern in EBV+ve cohort compared to the EBV–ve counterpart (Supplementary Fig. 22). Intriguingly, although NPC53 was identified with genetic mutations in *CYLD* and *TRAF3*, which is a genetic signature for EBV+ve NPC, the transcriptome profile of NPC53 revealed an inactivated NF-κB signaling and clustered with HK1 and NPC38. Given the reported NF-κB activating functions mediated by LMP1 (latent membrane protein 1) and

EBERs, it is postulated that loss of EBV episomes and its encoded RNAs and protein in NPC53 cells might lead to the decreased activity of NF-κB pathway[25–27]. Besides NF-κB and PI3K, upregulated gene expression in epithelial-mesenchymal transition, Notch and Wnt signaling pathways in EBV+ve cohorts was also revealed by GSEA (Supplementary Fig. 21b–d), indicating enhanced gene expression in these pathways might play functional roles in NPC carcinogenesis.

In summary, the newly established NPC PDXs and cell lines harbor common mutations present in the original patient tumors, and share similar signaling properties to NPC in patients, which supports their potentials for use in NPC research and preclinical drug evaluation.

**Phylogenetic study of EBV sequences in NPC PDXs/cell lines**. The sequencing reads of WGS mapped to mouse genome (mm10) and human genome (hg19) were removed and then aligned to the reference EBV genome (NC_007605). The mapped reads were used for de novo assembly of EBV genome sequences. Detailed summaries of sequencing data mapping and assembly are included in Supplementary Tables 5 and 6. The phylogenetic trees of EBV whole-genome and genes were constructed accordingly. As shown in Fig. 9, AG876, a type II EBV strain, clearly segregated from the other type I EBV strains, which is consistent with previous studies[28]. The whole-genome sequences of EBV in the newly established NPC PDXs (Xeno23, 32, 47 and 76) and cell line (NPC43) clustered with the Asian EBV strains, especially those sequences from Chinese NPC (M81, C666-1, HKNPC1-9 and GD2). The sequences of latent EBV genes, including *LMP1* and *EBNA1*, were subjected to phylogenetic analysis. The *LMP1* and *EBNA1* sequences from NPC EBV also clustered as a distinct category (Supplementary Fig. 23). It remains to be determined whether the clustering of NPC EBV strains may reflect an uneven geographical distribution of EBV strains or whether EBV retained in NPC may possess distinct biological property contributing to NPC pathogenesis.

## Discussion

Limited availability of representative NPC PDXs and cell lines has hampered both basic and translational research of NPC. The establishment and detailed characterization of new NPC PDXs, which recapitulate the mutational profiles as the original NPC in patients, will serve as useful preclinical NPC models for drug evaluation and facilitate the development of precision medicine for NPC treatment.

A distinct difference observed between the well-established NPC PDXs, which have been passaged for more than 25 years, and the newly established ones is the expression profile of EBV

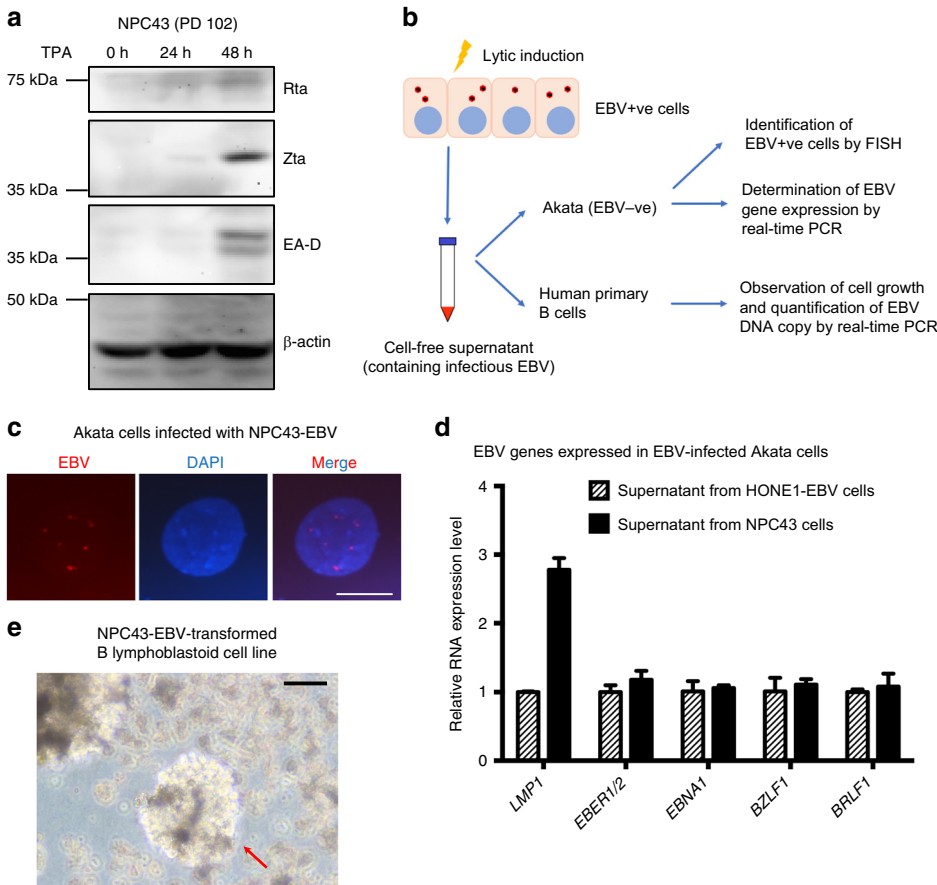

**Fig. 5** Production of infectious virus by NPC43 upon EBV lytic reactivation. **a** Expression kinetics of Rta, Zta and EA-D proteins in NPC43 cells upon TPA treatment. Expression of β-actin was included as the loading control. **b** The schematic diagram showing the workflow to determine whether infectious EBV particles were present in the supernatant of EBV+ve cells (including NPC43 and HONE1-EBV) upon lytic induction. Briefly the EBV+ve cells were induced to undergo lytic EBV reactivation. The cell-free supernatant which contains the infectious EBV was collected and was used to infect EBV−ve Akata cells or human primary B cells. The presence of EBV virions was confirmed by the respective experiments listed. **c** Identification of EBV+ve Akata cells by FISH analysis. The supernatant of NPC43 upon EBV lytic induction was collected for co-culture with EBV−ve Akata cells. At 96 h after co-culture, part of the infected Akata cells were subjected to FISH analysis. The presence of EBV genome was indicated by punctate red dots in Akata cell nucleus. Scale bar, 10 μm. **d** Determination of EBV gene expression by real-time PCR analysis in EBV-infected Akata cells. The supernatant collected from NPC43 or HONE1-EBV cells were subjected to co-culture with EBV−ve Akata cells for 96 h. EBV gene expression was evaluated afterwards. Akata cells infected by EBV from NPC43 cells after TPA treatment had a comparable level of EBV RNA expression with that from HONE1-EBV cells. Data are shown as mean ± SD from three independent experiments. **e** Proliferating foci of human primary B cells infected by NPC43-EBV after 28-day culture. Arrow suggests an NPC43-EBV-transformed B lymphoblastoid cell line. Scale bar, 100 μm

genes. Lytic EBV genes could be readily detected in the newly established PDXs compared to the long-term passaged ones. The long-term passage of NPC xenografts appears to select for latent EBV-infected NPC populations. Our study suggested that lytic EBV reactivation in NPC xenografts transplanted to immunodeficient mice may be an underlying reason interfering with the successful establishment of NPC PDXs. Presumably the intact tumor microenvironment in NPC plays an undefined but important role to support EBV latency, which is the predominant mode of EBV infection in NPC. The expression of latent EBV genes in NPC in patients may further contribute to NPC cell growth in patients through immune evasion and suppression of apoptosis[14]. Various cellular components and cytokines present in the NPC microenvironment may support latent EBV infection[29]. The transfer of NPC tissues to immune-suppressed mice devoid of human stroma may trigger lytic EBV reactivation. Hence, the inflammatory NPC stroma may represent an effective target for NPC treatment by disrupting the latency of EBV infection in NPC cells into lytic infection, which triggers tumor cell death and host immune response. In this study, we observed

that a long period was required for the NPC xenografts transplanted underneath the kidney capsule before being established as transplantable PDXs. This may reflect an adaptation of EBV-infected NPC cells in the xenografts to growth conditions in immunodeficient animals, and a selection of cells with less dependency on the NPC stroma. While latent EBV infection in NPC is the predominant mode, a low level of lytic EBV expression is nonetheless observed in small fractions of NPC cells in patient tumors[30]. The significance of expression of lytic genes in NPC is unclear but has been postulated to be involved in immune evasion[31]. An intricate balance of latent and lytic EBV gene expression may be involved in the maintenance of EBV in NPC cells and the survival of NPC cells.

The successful establishment of the EBV+ve NPC43 cell line from patient specimen by including ROCK inhibitor (Y-27632) in culture supports the hypothesis that suppression of lytic reactivation of EBV in NPC cells is crucial for EBV+ve cell line establishment, at least in culture condition. Recently, we have also established another EBV+ve cell line from C17 NPC xenograft using a similar approach[16]. C666-1 established from NPC

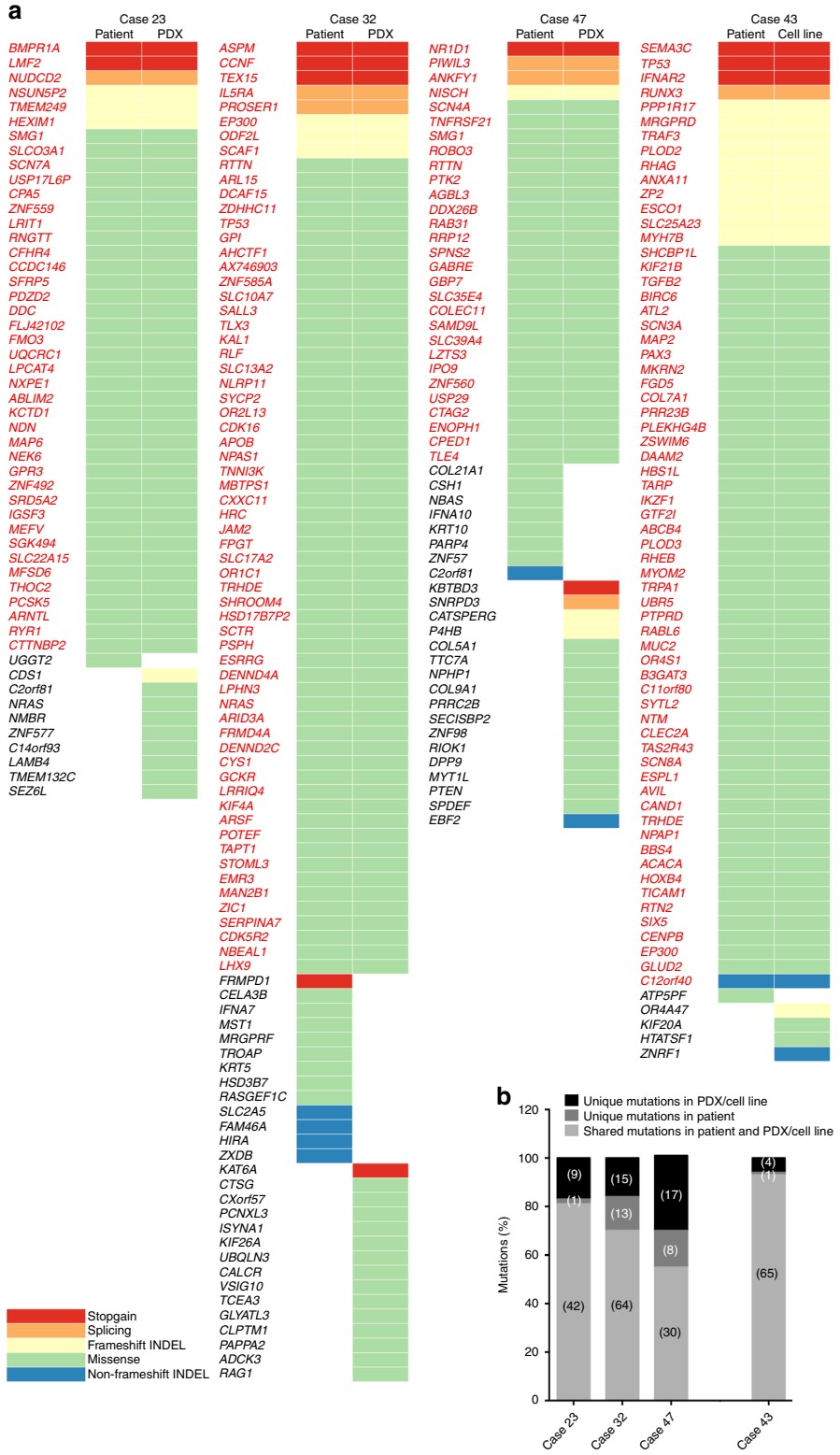

xenograft (X666) has been used extensively in investigations[4]. The C666-1, however, is defective in undergoing productive lytic EBV infection upon treatment with TPA and NaBu, or ectopic overexpression of *BZLF1* gene. The underlying reasons are unclear but may involve epigenetic regulation and mutation of genes involved in lytic reactivation of EBV in C666-1 cells[32]. Establishment of new and representative EBV+ve NPC cell lines is eminently required for NPC and EBV studies. The capacity of

NPC43 to undergo lytic reactivation of EBV and produce infectious EBV virions further makes it particularly useful for investigations of regulation of lytic and latent EBV infection in NPC. The detailed mechanisms of how Y-27632 suppresses EBV lytic reactivation in NPC43 require further investigations. Suppression of differentiation by Y-27632 may be essential for the establishment of EBV latency in NPC cells[11]. The hypothesis regarding the requirement of undifferentiated status in epithelial cells to

**Fig. 6** Somatic mutations in PDXs/cell line and their corresponding NPC tumors. **a** Mutation profiles of Xeno23, 32, 47 and NPC43 and their corresponding patient tumors. Somatic mutations, including stopgain (red box), splicing (orange box), frameshift INDEL (yellow box), missense (green box) and non-frameshift INDEL (blue box), were identified by WES analysis and plotted. For each mutation plot, the left column represents mutations in patient tumor, while the right column represents those identified in each newly established PDX or cell line. The mutated gene names are labeled at the left side of each mutation plot. The shared mutations between PDX/cell line and its corresponding patient tumor are labeled in red, while the uniquely mutated gene names in either PDX/cell line or its corresponding patient tumor are labeled in black. **b** Bar chart illustrating the percentage of gene mutations identified in each NPC case. Gray bars indicate the shared somatic gene mutations between PDX/cell line and its corresponding patient tumor; dark gray bars reveal the unique gene mutation(s) identified in NPC patient tumor; black bars indicate the unique gene mutations found only in the newly established PDX/cell line. The numbers with brackets labeled in each bar represent the number of mutations identified for each case. The overlap of mutations between NPC PDXs/cell line and their corresponding patient NPC tumors was 82% in Xeno23, 81% in Xeno32, 64% in Xeno47, and 94% in NPC43 cell line

establish EBV latency is also supported by the universal presence of EBV infection in the undifferentiated type of NPC prevalent in endemic areas including southern China, but absence in squamous carcinoma of head and neck cancer outside the nasopharynx in the same locality. Expression of latent EBV genes, notably *LMP1* and *BART*-microRNAs, has been postulated to support the growth of EBV-infected NPC cells[33]. Hence, modulating the differentiation and cell signaling properties in EBV-infected NPC to disrupt EBV latency may be of therapeutic potentials for NPC treatment[34]. In addition to the establishment of EBV+ve NPC cell line, NPC43, we have also established two EBV−ve NPC cell lines, NPC53 and NPC38, as the new cell line resources for NPC and EBV research. Both NPC38 and 53 will serve as useful EBV−ve NPC cell models in basic and preclinical studies in NPC.

Significant similarities of mutation profiles were revealed between PDX/cell line and patient NPC by WES data analysis. However, unique mutations were also identified in either PDX/cell line or patient tissue, which might be contributed by intratumor heterogeneity. Besides, the continuous selective pressure during PDX and cell line establishment could also be deposed for the dominant growth of some specific subpopulation of NPC cells. Distinct mutation profiles between EBV+ve and EBV−ve NPC characterized by WGS may provide hints to further study the requirements for stable EBV latent infection in epithelial cells, as well as its contribution to NPC carcinogenesis. Transcriptome analysis between EBV+ve and EBV−ve PDXs/cells suggest differential gene expression patterns in multiple signaling pathways. It remains to be determined whether the variation of these aberrant pathways is due to EBV infection, or a critical factor contributing to the latency and persistence of EBV in epithelial cells. Notably, the inactivated NF-κB signaling in NPC53 cells by transcriptome characterization suggests the important roles of EBV infection and its encoded genes in driving NF-κB signaling, and indicates that mutations in *TRAF3* and *CYLD* could be essential but not sufficient for the induction of potent NF-κB signals in NPC cells.

In summary, the full characterization of new NPC PDXs and cell lines will provide valuable resources for NPC and EBV research. The experience and knowledge gained from this study will also contribute to the future success in the establishment of more representative NPC models, which are important for understanding the properties of NPC and the roles of EBV in NPC pathogenesis.

## Methods

**NPC specimens**. NPC biopsies and nasopharyngectomized tissues used in this study were from patients admitted to Queen Mary Hospital, the University of Hong Kong, Hong Kong. The collection and use of these NPC specimens for this experimental study were approved by the Institutional Review Board of the University of Hong Kong, and the patients' consents were obtained. Tissues collected from patients were immediately immersed in M199 medium (Sigma-Aldrich) to maximize the viability of cells. Generally, the sample was washed and processed into 1 mm³ pieces in biosafety cabinet for surgical implantation in mice for PDX

establishment and/or explantation to primary culture for cell line establishment. Extra sample if available was fixed and subjected to histological examination.

**Surgical implantation to establish PDXs**. All animal care and experimental procedures were approved by the Committee on the Use of Live Animals in Teaching and Research, the University of Hong Kong. For implantation to subrenal capsule sites of NOD/SCID mice, the following surgical procedures were performed. A small skin incision was made along the dorsal midline of an anesthetized mouse. The kidney was then slipped out of the body cavity. A 2-mm incision was made in the kidney capsule. The open edge of the renal capsule was lifted and 2–3 pieces of NPC tissues (1 mm³) were carefully inserted into the subcapsular space. The kidney was gently inserted back into the body cavity. The body wall and skin were closed by sutures. For subcutaneous implantation, a skin incision was introduced dorsally at the franks of the mice to insert the explanted xenograft.

**Establishment of cell lines from NPC**. Small pieces of tumors (<1 mm³ in size) were explanted onto culture flasks with 2 ml of RPMI-1640 medium (Sigma-Aldrich) containing 10% FBS (Gibco), 100 U ml⁻¹ penicillin, 100 μg ml⁻¹ streptomycin and 4 μM Y-27632 (Enzo Life Sciences). The explant culture was maintained at 37 °C with 5% $CO_2$ in humidified air. After 3 days, 1 ml medium was added to the culture to avoid drying up of tumor tissues. Outgrowth of fibroblasts from explants was carefully removed using fire-polished ends of glass pipettes under an inverted microscope (Olympus). This process was carried out routinely (once or twice a week), depending on the growth rate of fibroblasts. Epithelial outgrowth migrating out from the explanted tumor tissues were scraped free of fibroblasts at the growth edge, and allowed to grow to near confluence in the culture flask before subculture. The epithelial cells were gently trypsinized to dissociate cells from the culture flask. For the first subculture, the dissociated cells were re-seeded onto the original culture flasks. Subculture was performed again to a new culture flask when the culture became confluent. Proliferation of cells was determined by direct cell counting and by ³H-thymidine incorporation[35].

A detailed description on the establishment of NPC43 has been included in the Results section. The presence of Y-27632 is essential for the expansion of epithelial outgrowths and continuous growth of cells from the NPC explant. Abrupt withdrawal of Y-27632 in NPC43 at early passages induced massive cell death. To examine effect of different concentrations of Y-27632 on the EBV gene expression of NPC43 cells at early passages, a stepwise strategy was used to decrease the concentration of Y-27632 in the culture medium. Briefly, NPC43 at PD 10 was maintained in medium containing 4 μM Y-27632 in the first week after subculture. Then, in the second week, the concentration of Y-27632 was reduced to 2 μM, and further to 1 μM in the third week. At the fourth week, the cells were eventually maintained in the medium without Y-27632 and harvested for the following experiments before they reached confluency. For the treatment of NPC43 cells with higher Y-27632 concentrations (10 μM and 20 μM), generally, the cells were maintained at the respective concentrations for 4 weeks. Establishment of cells using higher Y-27632 concentrations was not preferred as the fibroblasts would also have a higher chance to be immortalized, which may dominate the culture. The tumorigenicity of NPC43 cells was confirmed by subcutaneous injection in NOD/SCID mice. Around 10 million cells in 100 μl medium were mixed with an equal volume of Matrigel (BD Biosciences) and injected subcutaneously into the flanks of NOD/SCID mouse.

**Cell culture**. EBV−ve Akata, Namalwa, C666-1, HONE1-EBV and HK1 cells were maintained in RPMI-1640 medium supplemented with 10% FBS, 100 U ml⁻¹ penicillin and 100 μg ml⁻¹ streptomycin. C17 was maintained in the culture medium as above with additional supplemented Y-27632 at 4 μM. NP69 was maintained in Keratinocyte-SFM supplemented with human recombinant epidermal growth factor 1-53 and bovine pituitary extract (ThermoFisher Scientific). Briefly, Namalwa is a human lymphoblastoid cell line with 2 EBV genome copies integrated into the host genome[36]. EBV−ve Akata and Namalwa cell lines were kindly provided by Professor Kenzo Takada (Hokkaido University, Japan). The HONE1-EBV, C17 and NP69 cells were established in our laboratory[16,37,38]. C666-1 and HK1 cell lines were kindly provided by Professor Dolly Huang (Chinese

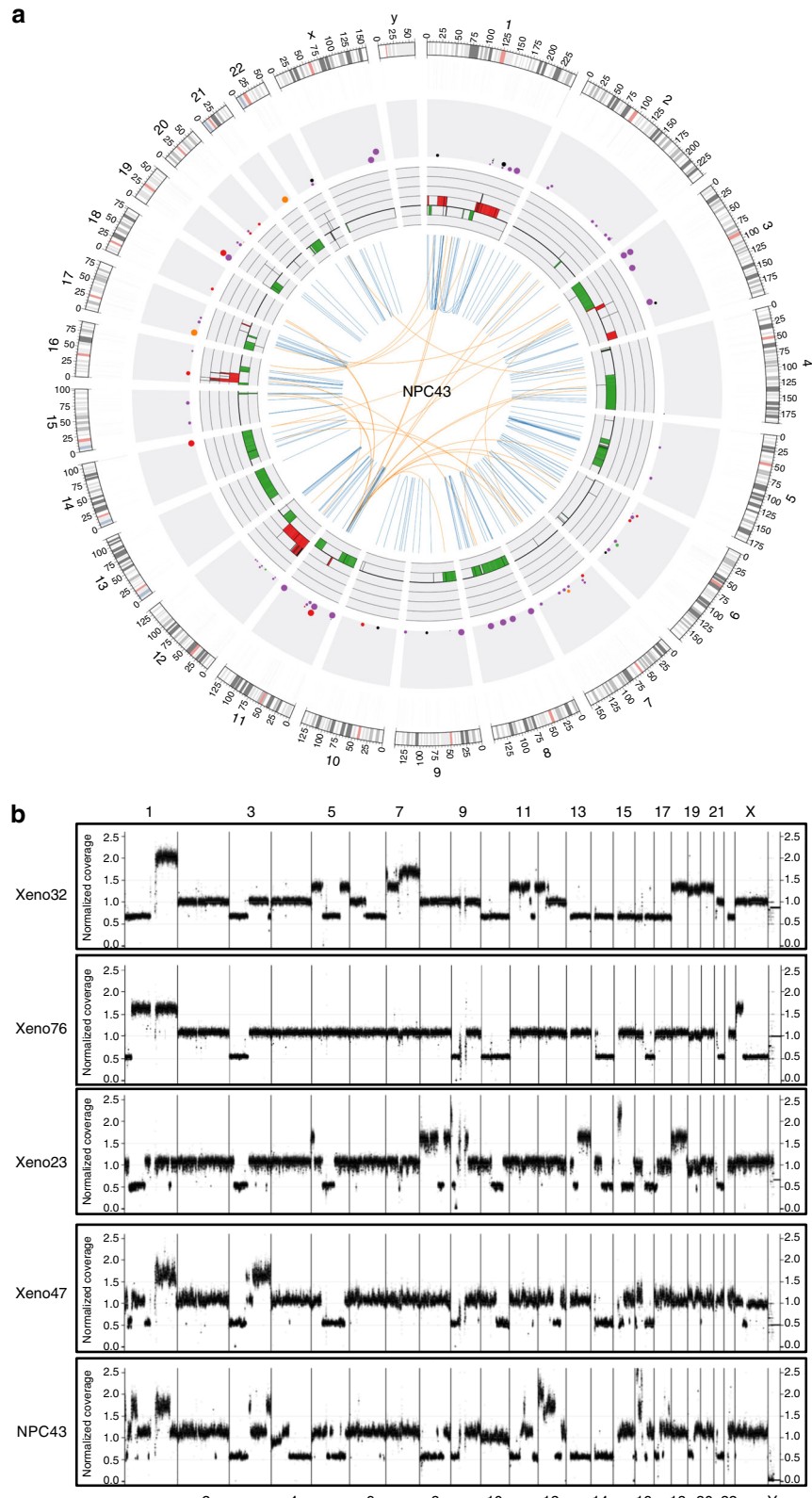

**Fig. 7** Genetic landscape of newly established NPC tumor lines. **a** Circos plot showing genomic aberrations in NPC43 by WGS. Illustrations in order from inner to outer rings: (1) structural variants (orange, inter-chromosomal rearrangement; blue, intra-chromosomal rearrangement), (2) copy number alterations (green, copy number loss; red, copy number gain; range: −2 to +4), (3) non-synonymous single-nucleotide variants or small indels (purple, missense; black, splicing; orange, nonsense; red, frameshift; green, inframe indel; brown, others) with allele frequency indicated by the size of each dot (0, 20, 40, 60, 80, 100% or more), (4) density of SNVs/small indels (0, 20, 40, 60, 80, 100, 120, 140, 160 per Mbps or more), (5) chromosome scale at 1 Mbps (shades of gray, cytobands; red, centromere). High-resolution plot is illustrated in Supplementary Fig. 14. **b** Whole-genome profiles of chromosome copy number alterations in NPC PDXs/cell line. Overall normalized coverage across genome is shown on vertical axis

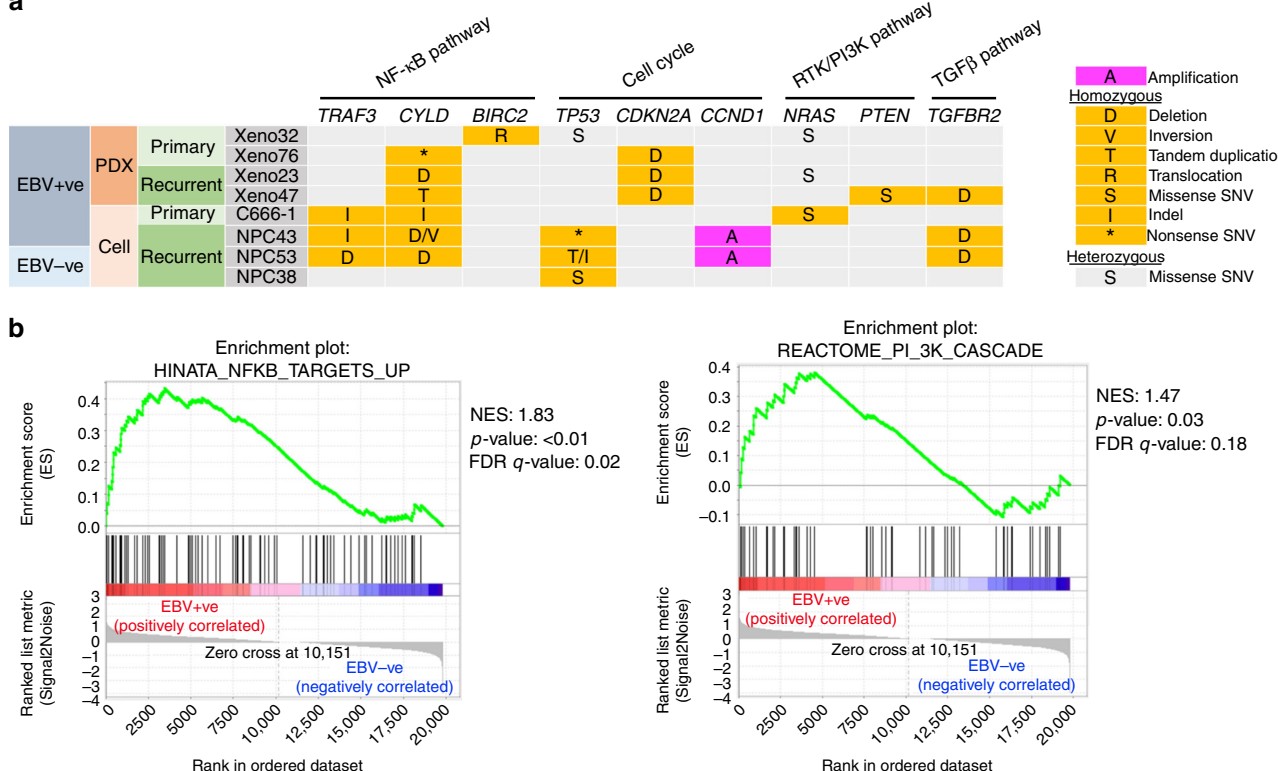

**Fig. 8** Mutations of key oncogenic pathways in NPC tumor lines. **a** Genetic landscape in NPC PDXs and cell lines. Somatic mutations in the newly established PDXs/cell lines and C666-1 are illustrated in mutation plot. Gray box with character inside indicates heterozygous mutation, while orange box with character(s) inside indicates homozygous mutation. S, missense single-nucleotide variant (SNV); D, deletion; V, inversion; T, tandem duplication; I, indel; *, nonsense SNV. **b** Activation of NF-κB and PI3K pathways in EBV+ve NPC xenografts and cell lines by transcriptome study. Gene expression levels were quantified by RNA sequencing and subjected to GSEA for pathway enrichment analysis. Enrichment score (ES) indicates the degree to which the specified gene sets are overrepresented at the top/bottom of a ranked list of total protein-coding genes in individual PDX/cell line. Normalized enrichment score (NES) computes density of modified genes in the dataset with random expectancies, normalized by gene numbers found in each gene cluster. False discovery rate (FDR) is calculated by comparing actual data with 1000 Monte-Carlo simulations. For each enrichment plot, top portion: running ES for the specified gene set; middle portion: where the member of gene set appears in the ranked list of all protein-coding genes; bottom portion: value of the ranking metric as a measurement of gene's correlation with phenotype (EBV+ve or EBV−ve)

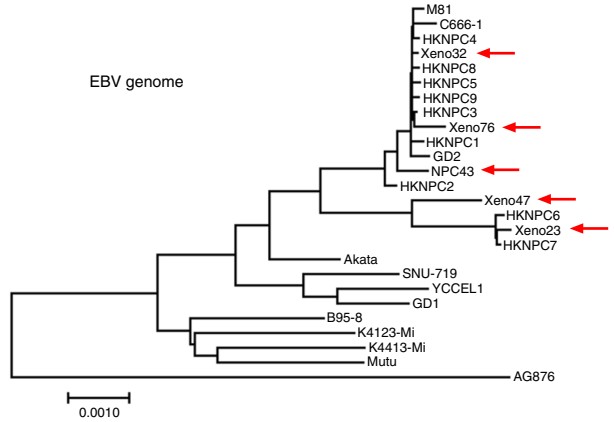

**Fig. 9** Phylogenetic study of whole EBV genomes in the newly established NPC models. De novo assembly of EBV whole-genome sequences was performed using WGS data. The assembled sequences were further subjected to phylogenetic analysis comparing with the currently publicly available EBV sequences. The phylogeny tree is drawn to scale, with branch lengths in the same units as those of the evolutionary distances used to infer the phylogenetic tree. The evolutionary distances were computed using the Maximum Composite Likelihood method. Arrows: EBV sequences in the newly established NPC PDXs and cell line, which show significant phylogenetic similarity with EBV sequences previously reported in NPC

University of Hong Kong)[4,39]. The cells were maintained at 37 °C with 5% $CO_2$ in humidified air.

**DNA extraction**. DNA from PDXs and cell lines was extracted using DNeasy® Blood&Tissue Kit (Qiagen) in accordance with the protocol recommended by the manufacturer. The purity and concentration of the extracted DNA were determined by NanoDrop2000 (ThermoFisher Scientific).

**Quantification of EBV copy number**. PCR amplification was carried out on MyiQ2 Two Color Real-Time PCR machine (BioRad). The primers and probes for *EBNA1* and *β-globin* were designed using the Universal Probe Library System (Roche Applied Science) (Supplementary Table 7). In each PCR reaction, the reaction mixture includes 5 μl DNA (10 ng μl⁻¹), 0.4 μl forward primer (10 μM), 0.4 μl reverse primer (10 μM), 10 μl LightCycler probe master mix, 4.05 μl PCR-graded water and 0.15 μl specific Universal Library probe. The reaction was initiated by pre-incubation at 95 °C for 10 min. Forty cycles of amplification were carried out by DNA denaturation at 95 °C for 10 s, annealing and elongation at 60 °C for 30 s. PCR was performed on serial dilutions of Namalwa DNA (harboring two EBV per genome) to generate two individual calibration curves for *EBNA1* and *β-globin*[40,41]. The average EBV copy number per cell was calculated according to the standard calibration curves prepared from Namalwa DNA.

**RNA extraction and quantification of EBV gene expression**. Extraction of total RNA and reverse transcription to cDNA were performed using TRIzol® reagent (Invitrogen) and SuperScript® First-Strand Synthesis System for RT-PCR (Invitrogen), respectively, according to the manufacturer's protocols[42]. Expression levels of EBV transcripts were examined by real-time PCR. The primers and probes for different genes were designed using Universal Probe Library System as listed in Supplementary Table 7. The expression levels of EBV genes were normalized to

*GAPDH* and the relative expression levels of genes of interest were determined by the $2^{-\Delta\Delta Ct}$ method.

**EBV FISH**. Harvested cells were treated with 2 ml 0.8% sodium citrate for 15 min at 37 °C, 20 μl 1:3 acetic acid/methanol (fixative solution) for 5 min at 37 °C and centrifuged at $115 \times g$ for 5 min at room temperature (RT). After removing supernatant, 5 ml fixative solution was added, followed by centrifugation at $115 \times g$ for 5 min, supernatant removal and addition of 5 ml fixative solution. This washing step was repeated 3 times, before spreading the cells onto the slide, which was air-dried. The slide was then aged at RT for 5–7 days before FISH. The aged slide was treated with 0.1 mg ml$^{-1}$ RNase A (DNase inactivated) for 1 h at 37 °C, 2× SSC (0.30 M sodium chloride, 0.03 M sodium citrate) for 10 min at RT, 0.015 μg ml$^{-1}$ proteinase K for 15 min at 37 °C, fixed with 3% paraformaldehyde for 10 min at RT, and washed with 2× SSC for 10 min at RT. The slide was dehydrated with 70%, 85% and 95% ethanol for 2 min each at RT and air-dried with nitrogen gas. The biotin-labeled probe targeting EBV *Bam*HI-W repeats (kindly provided by Professor Bill Sugden, University of Wisconsin-Madison, USA) dissolved in hybridization solution (Cytocell) was denatured for 5 min at 80 °C, and incubated for 30 min at 37 °C. The slide was placed into denaturing solution (70% formamide dissolved in 20× SSC) for 4 min at 80 °C, dehydrated with 70%, 85% and 95% ethanol for 2 min each at RT and air-dried. Then, the probed was added onto the slide and covered with a coverslip, which was then sealed with rubber cement. The slide was incubated overnight at 37 °C in a humidified chamber, washed with 50% formamide for 5 min twice at 45 °C and 2× SSC for 5 min twice at 45 °C. Streptavidin-labeled Cy3 (Sigma-Aldrich) was added onto the slide, which was then incubated at RT for 40 min, washed with 2× SSC for 5 min twice at 45 °C, dehydrated with 70%, 85% and 95% ethanol for 2 min each at RT, and air-dried with nitrogen gas. DAPI (4′,6-diamidino-2-phenylindole) was added onto the slide for DNA staining. The slide was covered with a coverslip. Fluorescence images were captured under a Leica fluorescence microscope by a computer equipped with SPOT software (Leica).

**Spectral karyotyping analysis**. Cells were treated with 0.03 μg ml$^{-1}$ colcemid (Sigma-Aldrich) for 6 h before harvest. The cell spreading, slide preparation and treatments were performed as the same as described above for FISH. The 24-color SKYPaint probe (Applied Spectral Imaging) was denatured for 7 min at 80 °C and incubated for 30 min at 37 °C. After slide incubation with SKY probe, the slide washing and staining were carried out in accordance with the protocols provided by the manufacturer. Spectral karyotyping images were acquired using the SkyVision Imaging System equipped with a Zeiss Axioplan 2 fluorescence microscope. Karyotyping was performed using SKY View 2.0 software (Applied Spectral Imaging)[43].

**Histological characterization**. Tumor samples from patients and mice were fixed in 10% neutral buffered formalin. Paraffin blocks were prepared and serial 5-μm-thick sections were cut from paraffin-embedded tumors. Consecutive sections of PDXs were used in the H&E staining, *EBER* ISH and immunohistochemical analysis of cytokeratin AE1/AE3. *EBER* ISH staining was performed using EBV probe in situ hybridization kit (Novocastra) according to the manufacturer's instructions[44]. For immunohistochemical staining against cytokeratin AE1/AE3, the paraffin sections were de-paraffinized and rehydrated for subsequent staining. Following antigen retrieval, endogenous biotin activity was blocked by normal bovine serum and the sections were incubated with primary antibody (1:50; Dako, #M3515) in a moist chamber. Horseradish peroxidase-conjugated secondary antibody (Dako, #K4001) was applied to the sections, followed by incubation of DAB (3,3'-diamino-benzidine; Dako) substrate for color development. The slides were then dehydrated and mounted with Permount mounting medium (Fisher Scientific). The mRNA expression of EBV lytic genes in PDXs was detected by an RNAscope® 2.0 assay (Advanced Cell Diagnostics) with specific probes (*BZLF1*, *BRLF1*, *BMRF1* and *BLLF1*) according to the manufacturer's instructions[45]. C15 NPC xenograft and two EBV+ve NPC patient samples, NPC1 and NPC2, were also included in the panel for analysis and comparison. *EBER* staining was performed using RNAscope® specific probe of *EBER* in the two clinical specimens, and confirmed they are EBV+ve (Supplementary Fig. 24).

**Induction of lytic EBV replication**. Lytic EBV replication was induced in NPC43 and HONE1-EBV cells by different approaches. For early passages of NPC43, lytic EBV reactivation was induced by stepwise removal of Y-27632 as described in previous section or removal of Y-27632 for 48 h. Induction of lytic EBV reactivation in NPC43 could also be achieved by TPA treatment (40 ng ml$^{-1}$). In HONE1-EBV, EBV lytic reactivation was induced by the combined treatment with TPA and NaBu, or treatment with suberoylanilide hydroxamic acid (SAHA) alone, which were effective in induction of lytic infection of EBV[34]. Cells were harvested for western blot analysis, IF staining and real-time PCR after respective treatments.

**Detection of infectious EBV from NPC43 upon lytic induction**. Upon lytic reactivation of EBV in NPC43 cells by TPA treatment for 48 h, cells were rinsed twice with phosphate-buffered saline (PBS) and replenished with fresh medium. Fresh medium was used to culture the TPA-treated cells for 72 h to collect infectious viral particles in the supernatants. The harvested supernatant was centrifuged at $115 \times g$ for 5 min and then filtered through a cellulose acetate filter (0.45 μm) (Sartorius) to remove cell debris. Then, the supernatant was centrifuged at $37,500 \times g$ for 4 h at 4 °C to concentrate the viral particles. The supernatant was discarded, and the pellet was resuspended with fresh RPMI-1640 medium in 1/20 of its original volume. The medium containing viral particles was then used to infect EBV–ve Akata cells. The infected Akata cells were harvested and subjected to DNA extraction for EBV copy and RNA extraction to determine EBV gene expression by real-time PCR, and EBV DNA FISH for determination of EBV+ve cells.

**Primary B-cell transformation**. Peripheral blood mononuclear cells were resuspended in 3.6 ml pre-warmed culture medium, followed by the addition of cyclosporine A (Sigma-Aldrich) to a final concentration of 0.5 mg ml$^{-1}$. Then, 400 μl of 100× concentrated EBV from NPC43 was added into cell suspension. Aliquots of cell suspension were prepared into 4 wells in 24-well plate. Proliferating foci of transformed B cells could be observed 2–3 weeks after infection. Quantification of EBV DNA in transformed B cells was determined by PCR of EBV DNA polymerase gene for EBV viral genome copies[46]. The primers and probe for EBV DNA polymerase gene were: (forward) 5'-CTTTGGCGCGGATCCTC-3'; (reverse) 5'-AGTCCTTCTTGGCTAGTCTGTTGAC-3'; (FAM-labeled probe) 5'-CTTTGGCGCGGATCCTC-3' (Applied Biosystems). The PCR conditions were: 50 °C 2 min; 95 °C 10 min; 95 °C 15 s and 60 °C 60 s for 40 cycles.

**Western blots**. Cells were rinsed twice in PBS and lysed in ice-cold RIPA lysis buffer containing 50 mM Tris-HCl (pH 8.0), 150 mM NaCl, 1% Nonidet P40, 0.5% deoxy-cholic acid, 0.1% sodium dodecyl sulfate (SDS), protease and phosphatase inhibitors (1 mM phenylmethylsulfonyl fluoride (PMSF), 4 μg leupeptin, 4 μg apoptinin, 1 nM sodium fluoride and 1 nM sodium orthovanadate). Equal amounts of proteins from each sample were separated by SDS–polyacrylamide gel electrophoresis and the separated proteins were transferred to polyvinylidene fluoride membrane (Millipore). The chemiluminescence signal was captured using myECL Imager (ThermoFisher Scientific). β-Actin (1:2000; Santa Cruz, #sc-1616) was used as the loading control. The rabbit polyclonal BALF5 (1:1000) antibody was kindly provided by Professor Tatsuya Tsurumi (Aichi Cancer Center Research Institute, Japan). The antibodies against EA-D (1:500) and Gp350/220 (1:500) were obtained from Professor Jaap Middeldorp (Department of Pathology, Vrije Universiteit University Medical Center, Netherlands). Antibodies against Zta (1:1000, #11-007) and Rta (1:1000, #11-008) were purchased from Argene. Images of uncropped blots are shown in Supplementary Fig. 25.

**IF staining**. Cells were rinsed twice with PBS and treated with ice-cold pre-extraction buffer (pH 6.8) containing 10 mM HEPES, 100 mM NaCl, 300 mM sucrose, 1 mM magnesium chloride, 1 mM EGTA, 1 mM dithiothreitol, 1 mM PMSF, 10 μg ml$^{-1}$ aprotinin and 0.5% Triton X-100 for 2 min. The cells were then fixed with ice-cold pure methanol for 30 min and then incubated with blocking buffer (10% FBS and 0.2% Triton X in PBS) for 30 min. Primary antibodies against cytokeratin AE1/AE3 (1:100; Dako, #M3515), Zta (1:1000; Argene, #11-007), EA-D (1:500, a gift from Professor Jaap Middeldorp) and BALF2 (1:500) (a kind gift from Professor Tatsuya Tsurumi, Aichi Cancer Center Research Institute, Japan) were used. Alexa Fluor-conjugated secondary antibodies (1:500) were from Molecular Probes (#A11059 for cytokeratin AE1/AE3, Zta and EA-D and #A21206 for BALF2). The percentage of positive cells was quantified by calculating 10 microscopic views with over 2000 cells included.

**Transmission electron microscopy**. Routine transmission electron microscopy protocols were used to process PDXs harvested from mice. Briefly, fresh tissues were fixed in primary fixative containing 2% paraformaldehyde and 2.5% glutaraldehyde. Then tissues were post-fixed in 1% osmium tetroxide, followed by dehydration and embedding. Ultra-thin sections were prepared and stained with uranyl acetate, followed by staining with lead citrate. Philips CM100 transmission electron microscope was used to obtain images.

**WES**. For WES, 250 ng genomic DNA of xenograft samples, human tumors and blood samples were fragmented by an ultrasonicator (Covaris). These fragments were amplified using NEBNext UltraTM DNA library Prep Kit (NEB). The concentration of the libraries was quantified by a bioanalyzer (Agilent Technologies). The amplified fragments were hybridized to a TruSeq capture kit (Illumina) or SeqCap EZ kit (Roche) for enrichment; non-hybridized fragments were then washed away. The magnitude of the enrichment was estimated using real-time PCR. Paired-end, 100 bp (TruSeq) or 150 bp (SeqCap EZ) read-length sequencing was performed on the HiSeq 2000 sequencer according to the manufacturer's instructions (Illumina). Sequencing reads from xenograft samples were first mapped to two mouse reference genomes (mm10 downloaded from UCSC and NOD/ShiLtJ genome downloaded from Mouse Genomes Project, Sanger Institute) with Burrows–Wheeler Aligner (BWA) (0.7.17)[47]. About 23–39% of reads can be mapped to mouse genomes with mapping quality more than 15 in the xenograft samples. These reads were excluded from analysis to eliminate mouse sequence contamination. Then sequencing reads were aligned to the human genome (hg19). Picards were applied to sort output bam files and mark duplicates. GATK (3.8)[48] was applied for paired local realignment around INDELs, base quality recalibration,

variants discovery and quality control according to GATK Best Practices recommendations[49,50]. Somatic single-nucleotide polymorphisms and INDELs were called using MuTect (1.1.7)[51] and VarScan (2.3.7)[52], respectively. Somatic mutations were further filtered, if they are present in public databases (1000G and ESP6500) or in-house controls (>1000) with minor allele frequency more than 1%. To avoid the discrepancy caused by two capture kits, only the mutations with at least 15 reads for coverage in both TruSeq and SeqCap EZ kits were included in the analysis. All non-silent somatic mutations were then manually checked in Integrative Genomics Viewer (IGV, version 2.3, Broad Institute) to further remove variants of poor quality or mouse contamination.

**WGS**. For WGS, 1 μg of genomic DNA extracted from NPC cell lines, PDXs and matched normal samples were subjected to the Illumina Whole Genome Sequencing Service in Macrogen (Seoul, Korea). Standard Illumina protocols and Illumina paired-end adapters were used for library preparation from the fragmented genomic DNA. Sequencing libraries were constructed with 500 bp insert length. WGS was performed using the Illumina HiSeq 2000 platform with a standard 100 bp paired-end read[53]. Mean target coverage of 40× and 60× was achieved for the normal and tumor samples, respectively. The raw sequence reads were processed and aligned to hg19 human reference genome using Isaac aligner (01.15.02.08)[54]. Identification of somatic SNVs and SVs was conducted by Strelka (1.0.14)[51] and Manta (0.20.2)[55], respectively. We predicted somatic copy number aberration (CNA) and allelic imbalance in cancer genome using Patchwork-R (2.4)[56]. The identified somatic CNAs and SVs of each NPC were visualized by CIRCOS (0.69-4)[57].

**RNA sequencing**. Total RNA was extracted from NPC cell lines and xenografts using TRIzol® reagent (Invitrogen). RNA sequencing libraries were prepared by KAPA stranded mRNA-seq kit (Roche). Next-generation sequencing (100 bp, paired-end) using the Illumina HiSeq 1500 sequencing system was performed at Centre for Genomic Sciences, University of Hong Kong. Total sequencing reads were filtered for adapter sequence, low-quality sequence and ribosomal RNA sequence, and the reads were subjected to downstream data analysis. Briefly, the reads were mapped and aligned to human reference genome (hg38, Gencode) by STAR (2.5.2)[58]. Gene expression levels were quantified by RSEM (1.2.31)[59], and the differentially expressed genes between EBV−ve and EBV+ve samples were identified by EBSeq (1.10.0) with the criteria as false discovery rate (FDR) $q$-value below 0.05[60]. The expression levels of protein-coding genes were further subjected to GSEA version 3.0 to characterize the differences in transcriptome profiles of EBV+ve cohort in specific pathways as compared to the EBV−ve counterpart[61–63]. Heatmap was drawn by pheatmap R package (1.0.10; http://cran.r-project.org/web/packages/pheatmap/).

**Phylogenetic analysis of EBV genome sequences**. The non-human and non-mouse reads from WGS were aligned to the reference EBV genome (NC_007605) using BWA software[47]. The generated BAM files were subjected to SAMtools software (1.3)[64] for pile-up files and assessment of coverage of reads. The last 30 bases of the output reads were trimmed from the 3' ends of the aligned reads by the FastTrimmer of FASTX-Toolkit (0.0.13.2), while the first 70 bases from the 5' end were retained. After calculating the average coverage of reads, high-quality reads were assembled using the Velvet (1.2.07)[65]. The settings were optimized using the expected average $k$-mer coverage of 200 to 600, $k$-mer lengths of 35 and the minimum $k$-mer coverage of 20 to 70. The location and orientation of generated contigs by Velvet were examined by pairwise alignment to reference EBV genome (NC_007605). PCR primers were designed at the breakpoints between contigs. Sanger sequencing was performed to join the contigs. Multiple sequence alignment of all generated EBV genomes as well as publicly available ones were performed using MAFFT version 7[66]. The aligned sequences were visualized and edited using Jalview software (2.9.0b2)[67]. Poorly aligned regions were trimmed before construction of the phylogenetic tree. Phylogenetic analysis was performed using Molecular Evolutionary Genetics Analysis version 7 (MEGA7) by neighbor-joining algorithm[68]. In this study, multiple sequence alignments of EBV whole genomes or individual genes (including *LMP1* and *EBNA1*) were conducted in all sequenced EBV genomes for phylogenetic analysis.

**Statistical analysis**. All results were expressed as mean ± SD. Statistical analysis of imaging data quantification was performed using two-tailed $Z$-test, while other experimental data were statistically analyzed using two-tailed Student's $t$-test, and differences were considered significant at $p < 0.05$.

## Data availability

The WES and RNA sequencing data that support the findings of this study have been deposited in Sequence Read Archive (SRA) with accession numbers as SRP158745 and SRP158866, respectively. The WGS data have been deposited in European Nucleotide Archive (ENA) with accession number as PRJEB24495.

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

## Acknowledgements

The authors acknowledge the funding supports from Research Grant Council, Hong Kong: AoE NPC grant (AoE/M-06/08), Theme-based Research Scheme grant (T12-401/13-R) and Collaborative Research Fund (C7027-16G) and General Research Fund (17120814, 17161116, 17104617, 17114818, 17116416, 17111516, 17110315). The establishment and characterization of NPC xenografts were also supported by HMRF grant (04151726, 13142201). Imaging data were acquired using facilities in the Faculty Core Facility, Li Ka Shing Faculty of Medicine, The University of Hong Kong.

## Author contributions

W.L. and Y.L.Y. performed cell line establishment and characterization of their growth properties. W.L., Y.L.Y., Y.C.C., B.L. and C.K.C. contributed to establishment of PDXs. K. Y.Y., S.-D. L., K.W.L. and J.S.-H.K. performed WGS analysis. H.Z., W. Dai and J.M.Y.K. performed WES analysis. L.J. and W. Dai performed transcriptome analysis. L.J., K.F.H., A.K.S.C. and H.K. analyzed sequences of EBV genome. W. Deng, Y.Z. and T.L. performed EBV and cytogenetic analysis. G.T.Y.C. performed *EBER* and RNAscope® analysis. K.F.H. performed primary B-cell transformation assay. J.M.N. performed pathological examination of specimens. J.Y.-W.C., D.L.-W.K. and V.H.-F.L. provided samples and clinical information. H.C. detected EBV gene expression. P.M.H. and J.Z. performed lytic EBV induction. C.M.T. performed EBV infection. P.B., X.L., S.T.C., J.M. and A.L.-M.C. optimized protocols for establishment of xenograft and cell lines, and revised the manuscript. S.W.T., W.L., Y.L.Y., L.J., K.W.L. and M.L.L. wrote the paper.

## Additional information

**Competing interests:** The authors declare no competing interests.

