## [Peer Review File · Nature Communications]

Reviewers' Comments:

Reviewer #1:

Remarks to the Author:

This is a well-designed study in which the authors have successfully established the nasopharyngeal carcinoma (NPC) patient derived-xenograft (PDX) and patient derived-cell (PDC) models. In addition, the authors demonstrated the mutational profiles and genomic landscape of these NPC models. However, here are concerns regarding some of the author's interpretation of their data and some criticisms:

1. Please add more clinical information and follow-up information of the NPC patients involved in this study, such as EBV DNA copy number and EBV related Ab titer, treatments patients received and outcomes of these patients. Since it is suggested that the EBV lytic infection is related to the successful establishment of NPC PDX in your manuscript, whether there is any link between the patients' clinical information and successfully established PDX model. Please also discuss this. These information might be important for the future use of PDX model.
2. In figure 3, the authors use Y27632 to increase the successful rate of NPC PDC, whether the pre-treatment or treatment of Y27632 will also facilitate the establishment of NPC PDX.
3. In the current manuscript, the authors have conducted the mutational profiles and genomic landscape of these in vivo and in vitro NPC models, further indicating several signal pathways might involve in the pathogenesis of NPC. Whether is there genetic events between PDX from primary tissues and recurrent tissues.
4. It will be help to perform transcriptome assay of the NPC PDX43 and other successfully maintained PDX, the NPC EBV(+) PDC43 and NPC EBV(-) 38 and 53. Are there any important differences between these EBV(+) and EBV(-) samples? Whether the differences will help maintain EBV or be the outcomes of EBV existence? Whether the differences might potentially influence important cancer-associated pathways? Whether is there any significant pathway related to NPC recurrence by comparing the PDXs or cells from primary tissues and recurrent tissues.
5. The authors characterized genomic aberrations of the various NPC-PDX confirms these models have a striking molecular resemblance to their originating matched patient tissue. Why the PDX showed more mutation than the patient tissues did, is the selection of PDX or the intra-tumor heterogeneity? The authors need to comment the maintaining of the intra-tumor heterogeneity during the establishment of PDX or PDX.

Minor comments:

Please describe whether NPC 1 and 2 are EBV positive or not. Please describe them in the methods.

Reviewer #2:

Remarks to the Author:

This paper provides a comprehensive assessment of the establishment of a new panel of nasopharyngeal carcinoma (NPC) cells lines. It represents a significant contribution to the field in describing the full characterisation of these NPC xenografts and of an in vitro cell line (NPC43). Furthermore, this paper highlights the difficulties in establishing NPC cell lines and implicates the induction of the EBV lytic cycle as a major barrier to stable NPC cell line growth. The genetic characterisation of the NPC cell lines is described as well as a comprehensive comparison with the original primary tumours. This paper is an important addition to the literature in this field.

Specific minor comments.

1. The poor growth of NPC43 as a xenograft is intriguing given its ability to grow in vitro. What is the explanation for being able to propagate continuous cell lines from NPC38, 43 and 53 but not the other EBV-positive xenografts?
2. It is assumed that the continued culture of NPC43 requires constant exposure to the ROCK inhibitor. What happens when Y27632 is removed after passage 100?
3. The data on EBV gene expression in the NPC43 cell line should be included in the main paper not as a supplementary figure. It is important to present the full characterisation of this important cell line.
4. Gardella gels would be useful in determining the relative proportions of circular versus linear EBV genomes in the studies examining lytic reactivation in the NPC43 cell line.
5. The experiments rescuing EBV from NPC43 into EBV-ve Akata cells are interesting. Was a stable Akata cell line derived using the NPC43 virus? Does the EBV derived from NPC43 transform primary B cells?

6. Somatic mutations in the NPC43 cell line versus the original corresponding tumour are not presented? This might provide important insights.
7. There is a confusion in the paper relating to the homozygous deletion of TGFBR2 gene. The results indicate that this deletion is found in Xeno23 but this is not described in Figure 7a?
8. More detailed comparisons at the genetic level between the EBV-positive and EBV-negative NPC cell lines may reveal important differences that help to explain the requirements for stable EBV infection.

Print Email

Resend E-mail

Point-to-point response to reviewers' comments:

RE: NCOMMS-17-34317A

Title: *Establishment and characterization of new tumor xenografts and cancer cell lines from EBV-positive nasopharyngeal carcinoma*

Our replies to the comments raised by the Reviewers and changes made accordingly in this revision are elaborated as follows:

Reviewer #1's comments (Expertise: NPC, therapy, Remarks to the Author):

This is a well-designed study in which the authors have successfully established the nasopharyngeal carcinoma (NPC) patient derived-xenograft (PDX) and patient derived-cell (PDC) models. In addition, the authors demonstrated the mutational profiles and genomic landscape of these NPC models. However, here are concerns regarding some of the author's interpretation of their data and some criticisms:

- 1. Please add more clinical information and follow-up information of the NPC patients involved in this study, such as EBV DNA copy number and EBV related Ab titer, treatments patients received and outcomes of these patients. Since it suggested that the EBV lytic infection is related to the successful establishment of NPC PDX in your manuscript, whether there is any link between the patients' clinical information and successfully established PDX model. Please also discuss this. These information might be important for the future use of PDX model.*

Authors' response and follow-up: We agreed with the Reviewer's comments that clinical information is important for the future use of these PDX models. We have included in **Table 1** and **Supplementary Table 2** in our revised manuscript more relevant clinical and follow-up information for the NPC patients involved in this study, including the gender, age, histopathological properties, clinical stages (TNM), presence or absence of recurrence/metastasis, treatment regimens and clinical outcomes. The information of plasma EBV DNA copy number is included in **Supplementary Table 3**. As suggested by our reviewer, we have examined whether there is any correlation in the success rate of the establishment of PDXs/cell lines with tumor content in NPC tissues, plasma EBV DNA copy number or overall staging of NPC patients. As shown in the figures attached below, we were not able to demonstrate any significant correlation of PDX/cell line establishment with these parameters, including the number of EBV DNA copy in patient plasma. However, given the small sample size in this study, the results may not be conclusive. The data of the antibody titer against EBV antigen were not available for comparison. EBV serology test is not used in routine clinical practice for NPC patients and is replaced by the more sensitive assay of EBV copy number in blood. We postulated that the NPC stroma in patient that constitute the tumor microenvironment might be the most important factor

supporting the growth of EBV-infected NPC cells in patients. Explantation of NPC tissues from patients to immunodeficient mice or *in vitro* culture environment devoid of the special tumor microenvironment of NPC may trigger lytic reactivation of EBV in the infected NPC cells, and prevent their growth outside of NPC patient tumors. We observed high expression of lytic EBV genes in the newly established xenograft, which is uncommon in NPC specimens and long-time passaged NPC xenografts, which supports our hypothesis. Furthermore, suppression of lytic EBV reactivation in infected NPC cells by a ROCK inhibitor has facilitated the establishment of a new EBV+ve NPC cell line (NPC43) in this study. Further investigations to define the contributions of tumor microenvironment to the growth of NPC cells would contribute to future success of NPC cell line establishment, which is in much need in this research field.

Comparisons of success rate on PDX establishment based on (A) tumor content in tissues; (B) patient plasma EBV DNA copy number and (C) overall staging of NPC patients. The detailed clinical information has been listed in Table 1, Supplementary Table 2, and Supplementary Table 3. There is no significant correlation between these parameters and the success rate of PDX or cell line establishment. However, the sample size is small and the results may not be conclusive.

- In figure 3, the authors use Y27632 to increase the successful rate of NPC PDX, whether the pre-treatment or treatment of Y27632 will also facilitate the establishment of NPC PDX.*

Authors' response and follow-up: Thanks for the Reviewer's suggestion. To test the effects of pre-treatment or treatment with Y-27632 on the success rate of NPC PDX establishment *in vivo* will be extremely difficult, particularly for NPC, because of the small size of primary biopsies and nasopharyngectomized tissues available for investigation. We barely have sufficient tissues for explantation to the animals for PDX establishment and/or for *in vitro* cell line establishment. The low take rate and the long growth period required for PDX establishment prevent us from systematically examining the effects of treatment of Y-27632 *in vivo* for PDX establishment. Furthermore, the pharmacokinetic properties and toxicity of long-term injection of Y-27632 to immunodeficient mice are unknown. It

is also unclear how long the effects of Y-27632 pre-treatment of NPC specimens would last. There are also reports suggesting that inhibition of ROCK by Y-27632 or other small molecules may exhibit anti-tumor effects in pancreatic ductal adenocarcinoma and melanoma *in vivo*^{1,2}. Hence, the results of *in vivo* treatment of mice with Y-27632 may not be straightforward and may be difficult to be interpreted. A careful design and systematic study will be required to properly address the effects of Y-27632 *in vivo*. We attribute the low take rate of PDX establishment to the lack of specific tumor microenvironment to support the growth of NPC PDXs in immune suppressed animals. Our *in vitro* results suggest that the contribution of Y-27632 in the successful establishment of NPC cell lines is likely related to its inhibitory effects of EBV lytic reactivation in NPC cells. As discussed in the response to Comment 1, a systematic study to clarify the effects of Y-27632 on EBV-infected NPC cells may contribute to future success of NPC cell line establishment.

- 3. In the current manuscript, the authors have conducted the mutational profiles and genomic landscape of these in vivo and in vitro NPC models, further indicating several signal pathways might involve in the pathogenesis of NPC. Whether is there genetic events between PDX from primary tissues and recurrent tissues.***

Authors' response and follow-up: We have attempted to compare the genetic changes between PDXs/PDCs derived from primary tissues (Xeno32 and 76) and recurrent tissues (Xeno23, 47, NPC43, 38 and 53). However, we could not identify genes exclusively mutated in primary NPC-generated PDXs or recurrent NPC-generated ones. However, the comparison is limited by the small number of NPC PDXs/PDCs established from primary and recurrent tissues, and the results may not be conclusive.

However, WES and WGS analysis (**Figure 6** and **Figure 7**) revealed the presence of similar genetic aberrations and gene mutations to those identified in the recent reports on genomic landscape of human NPC, including *CYLD*, *TRAF3*, *N-ras* and *TP53*, which further supports the postulated driver functions of these genes in the development of NPC by providing growth advantages for NPC PDXs/PDCs (regardless of their primary or recurrent origins)^{3,4}.

- 4. It will be helpful to perform transcriptome assay of the NPC PDX43 and other successfully maintained PDX, the NPC EBV(+) PDX43 and NPC EBV(-) 38 and 53. Are there any important differences between these EBV(+) and EBV(-) samples? Whether the differences will help maintain EBV or be the outcomes of EBV existence? Whether the differences might potentially influence important cancer-associated pathways? Whether is there any significant pathway related to NPC recurrence by comparing the PDXs or cells from primary tissues and recurrent tissues.***

Authors' response and follow-up: We agreed with the Reviewer's recommendation and have performed detailed transcriptome analysis for NPC PDXs (Xeno23, 32, 47 and 76)

as well as EBV+ve and -ve NPC cell lines. As mentioned in the manuscript, the Xeno43 PDX failed to be passaged in immunodeficient animals, and its RNA was not available for investigation. Instead, the transcriptome analysis was performed in NPC43, which is an EBV+ve NPC cell line established directly from the surgical NPC43 tumor sample. Two EBV-ve NPC cell line established in this study, NPC38 and 53, as well as three other NPC cell lines, including HK1 (EBV-ve, derived from well-differentiated NPC⁵), C666-1 (EBV+ve, derived from X666⁶) and C17 (EBV+ve, recently established from an NPC xenograft⁷) were also included in the transcriptome analysis. This represents the most comprehensive comparison study between EBV+ve and EBV-ve NPC *in vitro* and *in vivo* models. The differentially expressed genes between EBV+ve and -ve cohorts were carefully analyzed and illustrated in **Supplementary Figure 20** and **Supplementary Figure 22** in the revised submission. The results from the transcriptome analysis are also included in the revised manuscript (**Results, Page 11**).

Using gene set enrichment analysis (GSEA), we observed significant upregulation in the transcripts of NF- κ B targets and PI3K cascade in the EBV+ve NPC PDXs and cell lines (**Figure 7d** and **Supplementary Figure 20d**), as compared to EBV-ve counterpart. These findings are consistent with the genetic mutation profiles identified in these NPC PDXs and cell lines, with loss-of-function mutations of negative regulators of NF- κ B pathway, as well as the gain-of-function mutations in PI3K pathway (described in the **Results**). Meanwhile, we also observed significant enrichment of the upregulated transcripts in NPC PDXs and EBV+ve NPC cell lines in additional cancer-associated pathways including Epithelial_Mesenchymal_Transition, Notch_Signaling and Wnt_Beta_Catenin_Signaling pathways (**Supplementary Figure 20a-c**), indicating the differences in transcriptome profiles between EBV+ve and EBV-ve NPC PDXs/cell lines. We also attempted to examine whether there is any specific pathway significantly related to NPC recurrence by comparing the transcriptome profiles of the PDXs/cells established from primary NPC tumors and recurrent ones. However, no obvious differences were noted in these limited number of cases (data not shown).

The transcriptome analysis data, together with WGS and WES data of the NPC PDXs and NPC cell lines will be deposited for public access upon the acceptance of this manuscript for publication. We believe that these new information will be useful for elucidation of molecular mechanisms involving in EBV maintenance in infected epithelial cells, as well as characterization of the effective targets for EBV-encoded genes. The two EBV-ve cell lines established in this study (NPC38 and NPC53) will serve as valuable *in vitro* cell models as *bona fide* EBV-ve NPC cell lines, which are rare in NPC research field.

5. ***The authors characterized genomic aberrations of the various NPC-PDX confirms these models have a striking molecular resemblance to their originating matched patient tissue. Why the PDX showed more mutation than the patient tissues did, is the selection of PDX or the intra-tumor heterogeneity? The authors need to comment the maintaining of the intra-tumor heterogeneity during the establishment of PDX or PDX.***

Authors' response and follow-up: As rightly pointed out by our reviewer, the NPC-PDXs exhibit striking molecular resemblance to their original patient NPC tissues. However, the NPC-PDXs also harbor genomic mutations which could not be detected in the original patient NPC. There are several explanations for these unique genomic mutations in NPC-PDXs. First of all, the sequencing depth of WES analysis may not be sufficient to detect all the mutations present in the subpopulations of patient NPC samples. Furthermore, NPC samples in patients are commonly infiltrated with non-tumor stromal cells (commonly 30 to 70%), which may further mask the presence of genomic mutations in a subpopulation of NPC cells. There is also the issue of tumor heterogeneity in NPC tissues. The NPC tissues used for genomic characterization may harbor the subpopulations that are different from that in the tissues used for NPC PDX establishment, which may contribute to the differences in genomic profiles of NPC PDXs and their original patient NPC. Another important contributing factor is that during the establishment of NPC PDXs, there may be selective pressure for some specific subpopulation of NPC cells which are less dependent on stromal components (tumor microenvironment). It is commonly observed that NPC PDX has a much lower infiltration of lymphocytes compared to patient NPC. Some NPC clones carrying specific genomic mutations with growth advantage in the patient NPC may be selected during the PDX establishment and its propagation in immunodeficient mice. The selection of NPC subclones adapting to *in vivo* passage in immunodeficient mice is believed to be a continuous process and will become stabilized at later passages of PDXs. The comparison of genomic profiles between patient NPC and their derived NPC PDX/cell line in this study were examined for the first time, which offers an opportunity to examine the evolution of tumor heterogeneity in NPC upon long-term propagation in immunodeficient animals. For this purpose, we have frozen down the representative NPC xenografts at different passages for future evaluation of tumor evolution and their response to drug treatment, which will further facilitate the study of evolution of genetic changes in NPC PDXs during passages. The implication of tumor heterogeneity of NPC in PDX establishment has been included in the **Discussion** of the revised manuscript (**Page 13, Line 38**).

Minor comments:

Please describe whether NPC 1 and 2 are EBV positive or not. Please describe them in the methods.

Authors' response and follow-up: We apologize for the not clearly described status of EBV infection in these NPC specimens in the original manuscript. In the revised manuscript, we have included the clinical diagnostic information of NPC1 and NPC2 as well as their EBV infection status in the **Methods** session (**Page 18, Line 1**). Both NPC1 and 2 are EBV-positive. The *EBER* staining results of NPC1 and 2 are included in the **Supplementary Figure 24**.

Reviewer #2's comments (Expertise: EBV, NPC, Remarks to the Author):

This paper provides a comprehensive assessment of the establishment of a new panel of nasopharyngeal carcinoma (NPC) cell lines. It represents a significant contribution to the field in describing the full characterisation of these NPC xenografts and of an in vitro cell line (NPC43). Furthermore, this paper highlights the difficulties in establishing NPC cell lines and implicates the induction of the EBV lytic cycle as a major barrier to stable NPC cell line growth. The genetic characterisation of the NPC cell lines is described as well as a comprehensive comparison with the original primary tumours. This paper is an important addition to the literature in this field.

Specific minor comments:

1. *The poor growth of NPC43 as a xenograft is intriguing given its ability to grow in vitro. What is the explanation for being able to propagate continuous cell lines from NPC38, 43 and 53 but not the other EBV-positive xenografts?*

Authors' response and follow-up: The poor growth of NPC43 as a xenograft is intriguing. It showed signs of minimal growth initially and was passaged under kidney capsule for several rounds with minimal or stagnant growth. Eventually the transplanted PDX was absorbed and disappeared in the transplanted animals at the 5th passage. The size of PDX43 was too small to be harvested for detailed histological or molecular examination. We do not have much information for the explanation of the failure of PDX43 to be passaged. One postulation is that lytic reactivation of EBV was activated in NPC43 after transplanted to immunodeficient animals. We postulate that the tumor microenvironment of NPC plays an essential role in supporting NPC growth and EBV latency in patients. In NPC patients, latent EBV infection in NPC cells and expression of latent EBV genes and BART-microRNAs, have been shown to promote immune evasion and inhibit apoptosis, hence providing selective advantage for EBV-infected NPC to grow in patients⁸. The rich stromal infiltration in patient NPC tissues may play an essential role to support latent infection of EBV and expression of latent EBV genes, to support the survival of EBV-infected NPC in patients. Upon transplantation to immunodeficient animals, this unique tumor microenvironment was lost, which further induced activation of EBV lytic infection in NPC43 cells and poor growth of NPC43 xenograft, leading to its failure to be propagated as a PDX. In culture, we were able to suppress the lytic reactivation of EBV in NPC43 cells by including ROCK inhibitor Y-27362. Enhanced expression of lytic EBV genes was also observed in NPC43 at early passage but subsided at late passages in culture suggesting the selection of latent EBV-infected NPC cells for propagation in culture.

We also attribute the difficulty in establishing EBV+ve NPC cell lines in culture to lytic reactivation of EBV in NPC cells explanted to culture. In contrast, establishment of EBV-ve NPC cell lines such as NPC38 and NPC53, were relatively easier compared to EBV+ve NPC cells as the growth inhibitory effects of lytic EBV replication are not present. The authors had experienced a much higher rate of establishment of EBV-ve cancer cell line from other regions of the head and neck (Tsao, SW. Unpublished observation). In

establishment EBV+ve NPC cell lines, the suboptimal growth conditions in culture may induce stress response of EBV in infected NPC cells, which may activate lytic EBV infection and inhibit growth and establishment of EBV+ve NPC cells in culture. This study showed that inhibition of lytic reactivation of EBV and the maintenance of NPC in an undifferentiated stage supported the establishment of EBV+ve NPC cell lines. Further elucidation of the unique factors in the tumor microenvironment in patient NPC will contribute to the understanding of growth requirement of EBV+ve NPC and enhance future success in establishment of representative NPC cell lines for investigation.

2. *It is assumed that the continued culture of NPC43 requires constant exposure to the ROCK inhibitor. What happens when Y27632 is removed after passage 100?*

Authors' response and follow-up: We have performed the suggested experiment by our reviewer. Y-27632 was removed from the culture medium and the changes in the expressions of latent and lytic EBV genes were examined by quantitative PCR analysis in NPC43 cells at earlier passage (population doubling (PD) 72) and late passage (PD 280). We observed enhanced expression of EBV lytic genes (*BZLF1*, *BRLF1*, *BMRF1* and *BLLF1*) in NPC43 at PD 72 upon removal of Y-27632. At later passage (PD280), there was no significant increase in lytic EBV gene expression (**Supplementary Figure 6**). The removal of Y-27632 exhibited no significant effects in the expression of EBV latent genes (*LMP1*, *EBNA1* and *EBER1/2*) in both PD 72 and PD 280 NPC43 cells. These results indicate that EBV in NPC43 at later passage is more “adapted” to latent infection and less dependent on Y-27632 to suppress its lytic EBV reactivation. Presumably, there may be epigenetic alterations in the EBV genome upon long-term propagation of NPC43 or possible acquisition of altered cell signaling in NPC43 cells supporting latent EBV infection. These results are included in **Supplementary Figure 6** in the revised submission.

3. *The data on EBV gene expression in the NPC43 cell line should be included in the main paper not as a supplementary figure. It is important to present the full characterization of this important cell line.*

Authors' response and follow-up: As suggested by our reviewer, we have moved the data on EBV gene expression in the NPC43 cell lines to the main paper. Please kindly refer to the revised **Figure 3f** for the EBV gene expression profiles in NPC43 cell line.

4. *Gardella gels would be useful in determining the relative proportions of circular versus linear EBV genomes in the studies examining lytic reactivation in the NPC43 cell line.*

Authors' response and follow-up: Unfortunately, our laboratory has no experience in Gardella gel analysis. There was also not sufficient time to set up an international collaborative study with oversea laboratories to run Gardella gels for us. Instead, we have performed the following experiments to address this issue. We have conducted immunofluorescence staining to determine the proportion of cells showing responsiveness to lytic induction. As shown in **Supplementary Figure 7**, 1 to 2% of NPC43 cells showed

positive immunostaining of EBV lytic proteins, including Zta (BZLF1), EA-D (BMRF1) and BALF2, upon the treatment with TPA for 48 hours. This indicates that only a small percentage of NPC43 cells could be induced to undergo lytic infection of EBV. The production of infectious EBV was further confirmed by infecting EBV-negative Akata cells and the detection of EBV genes expressed in the infected Akata cells after exposed to supernatant harvested from NPC43 cells.

5. *The experiments rescuing EBV from NPC43 into EBV-ve Akata cells are interesting. Was a stable Akata cell line derived using the NPC43 virus? Does the EBV derived from NPC43 transform primary B cells?*

Authors' response and follow-up: As suggested by the Reviewer, we are also very interested to understand the infection and transformation capacity of NPC43-EBV. Several experiments have been performed to address this important property, including: to examine the ability of NPC43-EBV to establish a stable EBV-infected Akata cell line, we harvested the infected Akata cells after co-culturing with NPC43-EBV for 48 hours and performed FISH analysis to determine the percentage of EBV+ve Akata cells. As shown in **Figure 5c**, successful infection of Akata cells by NPC43-EBV could be identified by the presence of EBV inside the cell nucleus as indicated by bright red dots. The percentage of EBV+ve Akata cells detected was around 2%. The pooled Akata cells upon NPC43-EBV infection were then subjected to single-cell sorting by BD-Influx cell sorter, and the EBV copy number was monitored in the successfully grown single-cell clones by qPCR. **Supplementary Figure 8** shows two representative Akata clones upon 29 and 55 days *in vitro* culture after cell sorting. The average EBV copy number on Day 29 was 24.095 ± 3.372 for clone 1, and 0.021 ± 0.004 for clone 2, respectively. Interestingly, we observed that the average EBV copy number in NPC43-EBV infected Akata clones decreased upon propagation *in vitro* (0.394 ± 0.064 for clone 1, 0.003 ± 0.002 for clone 2, 55 days post single-cell sorting), suggesting that EBV infection *per se* may not confer additional growth advantage to the proliferating Akata cells in culture. As there is no selection marker for NPC43-EBV, e.g. GFP tag or antibiotic resistance marker, it was difficult to enrich the percentage of EBV+ve Akata cells. We have also attempted to examine the transformation capacity of NPC43-EBV. Briefly, the concentrated NPC43-EBV supernatant was used to infect primary B lymphocytes in human blood (for detailed protocol, please kindly refer to the **Page 18, Line 31-38**). As shown in **Figure 5e**, the presence of transformed B cells was only observed in one of the NPC43-EBV treated B-cell cultures after 28-day culture, but not in control cultures without treated with NPC43-EBV. Presence of EBV could be detected in the transformed B cells by qPCR. Hence the rate of B cell transformation by NPC43 EBV is low, which may be due to the low titer of EBV produced by low percentage of NPC43 induced to undergo lytic replication.

6. *Somatic mutations in the NPC43 cell line versus the original corresponding tumour are not presented? This might provide important insights.*

Authors' response and follow-up: We were able to retrieve DNA from the original patient 43 NPC sample, and performed WES analysis to compare the genomic mutation in the original NPC43 patient with the established cell line at its early passage (PD 10). The analysis revealed a close similarity of genomic mutations between the patient NPC and its derived NPC cell line (94% similarity). The results are incorporated in **Page 9-10** and **Figure 6**.

7. *There is a confusion in the paper relating to the homozygous deletion of TGFBR2 gene. The results indicate that this deletion is found in Xeno23, but this is not described in Figure 7a?*

Authors' response and follow-up: We apologize for our not clearly described mutations in *TGFBR2* gene. The results have been amended by separating two copy number alterations identified. A unique 9q24 amplicon harboring *JAK2*, *CD274 (PDL1)* and *PDCD1LG2 (PDL2)* genes was found in Xeno23. Homozygous deletions of *TGFBR2* on 3p24 were detected in Xeno47 and NPC43. Please kindly refer to **Page 10, Line 24** for the revised text. The homozygous deletions of *TGFBR2* in Xeno47 and NPC43 could also be referred in the **Supplementary Data 3**.

8. *More detailed comparisons at the genetic level between the EBV-positive and EBV-negative NPC cell lines may reveal important differences that help to explain the requirements for stable EBV infection.*

Authors' response and follow-up: **Figure 7c** illustrates the common genetic alterations in the 4 newly established EBV+ve NPC PDX (Xeno23, Xeno32, Xeno 47, Xeno 76), one EBV+ve NPC cell line (NPC43), two NPC -ve NPC cell lines (NPC38 and NPC53), together with C666-1 cell line, an EBV+ve NPC cell line established in early study⁶. At a glance, the genetic alterations in NPC38 are distinct compared to C666-1, NPC43 and NPC53. It is not too surprising that NPC38 exhibits unique mutational profile, as it is likely to be derived from an independent EBV-ve squamous carcinoma induced by radiation. The NPC53 patient sample was shown to be *EBER*-positive but the EBV episomes were lost in the cell line during establishment. Hence it is not too surprising to observe that the profile of NPC53 genetic alterations is closer to EBV+ve NPC cell lines (further discussed below).

In response to the suggestion of Reviewer 1 (Comment 4), we have performed transcriptome analysis of EBV+ve and EBV-ve cell lines established in this study. The results are summarized in **Figure 7d** and **Supplementary Figure 20-22**. The transcriptome analysis shows similarity in downstream signaling events among EBV+ve NPC cell lines (and also EBV+ve NPC PDXs established), notably in the activation of NF- κ B signaling (**Supplementary Figure 22**), which is less prominent in EBV-ve cell lines (NPC38, NPC53 and HK1). We attributed the activation of NF- κ B pathway to the presence of EBV infection in EBV+ve NPC cell lines to EBV infection. Latent EBV-encoded gene such as LMP1 is well recognized to be a potent NF- κ B activator⁹ Abundant expression of

EBV dsRNA (EBV-encoded RNAs (EBERs) may activate NF- κ B pathway via toll-like receptor 3 (TLR3)/TRAF6/TAK1/TAB2 axis^{10,11}. Furthermore, we also observed enhanced activation signaling pathways involved in PI3K activation, epithelial-mesenchymal transition, Notch and Wnt Beta Catenin signaling in EBV+ve NPC cells. Interestingly, NPC53 reveals a closer similarity to EBV-ve NPC cell line in transcriptome analysis, while genetically recapitulates the mutation signature of EBV+ve NPC lines (with mutations in NF- κ B negative regulators, including *TRAF3* and *CYLD*). Given the fact that the NPC53 cell line was derived from an originally EBV+ve NPC sample, the presence or absence of EBV infection in NPC cell lines appear to have a major impact on its transcriptome property. It is postulated that EBERs-TLRs-TRAF6 or other EBV-mediated signaling are critical driving forces for constitutive activation of NF- κ B pathway in NPC cells. Extended study to examine differences in genetic and transcriptome profiles involving more NPC cell lines in future will provide important insights on the alterations of cell signaling in NPC cells mediated by EBV infection. The results from transcriptome analysis between EBV-ve and EBV+ve cohorts are incorporated in **Results (Page 11)**.

References:

1. Huang, G.X., *et al.* Up-regulation of Rho-associated kinase 1/2 by glucocorticoids promotes migration, invasion and metastasis of melanoma. *Cancer Lett* **410**, 1-11 (2017).
2. Rath, N., *et al.* Rho Kinase Inhibition by AT13148 Blocks Pancreatic Ductal Adenocarcinoma Invasion and Tumor Growth. *Cancer Res* **78**, 3321-3336 (2018).
3. Li, Y.Y., *et al.* Exome and genome sequencing of nasopharynx cancer identifies NF-kappaB pathway activating mutations. *Nat Commun* **8**, 14121 (2017).
4. Zheng, H., *et al.* Whole-exome sequencing identifies multiple loss-of-function mutations of NF-kappaB pathway regulators in nasopharyngeal carcinoma. *Proc Natl Acad Sci U S A* **113**, 11283-11288 (2016).
5. Huang, D.P., *et al.* Establishment of a cell line (NPC/HK1) from a differentiated squamous carcinoma of the nasopharynx. *Int J Cancer* **26**, 127-132 (1980).
6. Cheung, S.T., *et al.* Nasopharyngeal carcinoma cell line (C666-1) consistently harbouring Epstein-Barr virus. *Int J Cancer* **83**, 121-126 (1999).
7. Yip, Y.L., *et al.* Establishment of a nasopharyngeal carcinoma cell line capable of undergoing lytic Epstein-Barr virus reactivation. *Lab Invest* (2018).
8. Kang, D., Skalsky, R.L. & Cullen, B.R. EBV BART MicroRNAs Target Multiple Proapoptotic Cellular Genes to Promote Epithelial Cell Survival. *PLoS Pathog* **11**, e1004979 (2015).
9. Eliopoulos, A.G., *et al.* Epstein-Barr virus-encoded LMP1 and CD40 mediate IL-6 production in epithelial cells via an NF-kappaB pathway involving TNF receptor-associated factors. *Oncogene* **14**, 2899-2916 (1997).
10. Jiang, Z., Mak, T.W., Sen, G. & Li, X. Toll-like receptor 3-mediated activation of NF-kappaB and IRF3 diverges at Toll-IL-1 receptor domain-containing adapter inducing IFN-beta. *Proc Natl Acad Sci U S A* **101**, 3533-3538 (2004).
11. Li, Z., *et al.* EBV-encoded RNA via TLR3 induces inflammation in nasopharyngeal carcinoma. *Oncotarget* **6**, 24291-24303 (2015).

Reviewers' Comments:

Reviewer #1:

Remarks to the Author:

The authors has addressed all the reviewer's concerns and performed the additional experiments. Hope all the RNA-seq, WGS, and WES data will be access by public in time after acceptation of this manuscript. This reviewer has no further concerns.

Reviewer #2:

Remarks to the Author:

The point-to-point response of the authors to my original comments are satisfactory and revised manuscript addresses my original concerns.